# Resolving the Taxonomic Status of Potential Biocontrol Agents Belonging to the Neglected Genus *Lipolexis* Förster (Hymenoptera, Braconidae, Aphidiinae) with Descriptions of Six New Species [note 1]

**DOI:** 10.3390/insects11100667

**Published:** 2020-09-29

**Authors:** Korana Kocić, Andjeljko Petrović, Jelisaveta Čkrkić, Nickolas G. Kavallieratos, Ehsan Rakhshani, Judit Arnó, Yahana Aparicio, Paul D. N. Hebert, Željko Tomanović

**Affiliations:** 1Institute of Zoology, Faculty of Biology, University of Belgrade, Studentski trg 16, 11000 Belgrade, Serbia; korana.kocic@bio.bg.ac.rs (K.K.); andjeljko@bio.bg.ac.rs (A.P.); jckrkic@bio.bg.ac.rs (J.Č.); ztoman@bio.bg.ac.rs (Ž.T.); 2Laboratory of Agricultural Zoology and Entomology, Department of Crop Science, Agricultural University of Athens, 75 Iera Odos str., 11885 Athens, Greece; 3Department of Plant Protection, College of Agriculture, University of Zabol, Zabol 98615-538, Iran; rakhshani@uoz.ac.ir; 4IRTA Cabrils, 08348 Cabrils, Spain; judit.arno@irta.cat (J.A.); yahana_adm@hotmail.com (Y.A.); 5Centre for Biodiversity Genomics, Biodiversity Institute of Ontario, University of Guelph, 50 Stone Road, East Guelph, ON N1G 2W1, Canada; phebert@uoguelph.ca

**Keywords:** *Lipolexis*, phylogeny, Aphidiinae, DNA barcoding, species discovery, cryptic species

## Abstract

**Simple Summary:**

*Lipolexis* is small but widely distributed genus across Europe and Asia. Nevertheless, its taxonomic distinctiveness was subsequently questioned by some authors who considered it as a synonym of the genus *Diaeretus*. Although *Lipolexis* is widely distributed and one species (*Lipolexis oregmae* Gahan) is an important biological control agent, the last taxonomic study on it was conducted more than 50 years ago. Our study employs an integrative approach (morphology and molecular analysis (COI barcode region)), to examine *Lipolexis* specimens that were sampled worldwide, including specimens from BOLD database. It led to the description of six new species. Each of the new species possesses clear morphological characters that distinguishes it from its congeners. Our findings suggest that two groups can be differentiated within the genus—*oregmae* and *gracilis*. Furthermore, we present a key for the identification to all known *Lipolexis* species of the world.

**Abstract:**

*Lipolexis* is a small genus in the subfamily Aphidiinae represented by one species in Europe (*Lipolexis gracilis* Förster) and by four in Asia (*Lipolexis wuyiensis* Chen, *L. oregmae* Gahan, *L. myzakkaiae* Pramanik and Raychaudhuri and *L. pseudoscutellaris* Pramanik and Raychaudhuri). Although *L. oregmae* is employed in biological control programs against pest aphids, the last morphological study on the genus was completed over 50 years ago. This study employs an integrative approach (morphology and molecular analysis (COI barcode region)), to examine *Lipolexis* specimens that were sampled worldwide, including specimens from BOLD database. These results establish that two currently recognized species of *Lipolexis* (*L. gracilis*, *L. oregmae*) are actually a species complex and also reveal phylogenetic relationships within the genus. Six new species are described and a global key for the identification of *Lipolexis* species is provided.

## 1. Introduction

*Lipolexis*, a small genus in the subfamily Aphidiinae, is widely distributed across Europe and Asia. Like all members of this subfamily, species of *Lipolexis* are solitary koinobiont endoparasitoids of aphids (Aphididae). Förster (1862) [1] described it as monotypic with *Lipolexis gracilis* as the type species. However, its taxonomic distinctiveness was subsequently questioned by some authors who considered it as a synonym of the genus *Diaeretus* [2,3]. Starý [3] re-established *Lipolexis* on the basis of type material and specimens collected in former Czechoslovakia. A second member of this genus, *Lipolexis oregmae* Gahan, was first described as *Diaeretus oregmae* by Gahan [4]. However, in his revision of generic boundaries in the family Aphidiidae (now subfamily Aphidiinae), Starý [5] stated that, aside from its type species, the genus *Diaeretus* was monotypic and other taxa should be transferred to other genera. Consequently, he reassigned *D. oregmae* to *Lipolexis* [6]. Two years later, Mackauer [7] described the third species, *Lipolexis scutellaris* Mackauer, from the Oriental region. However, after re-examining *L. oregmae* and *L. scutellaris*, Starý concluded that *L. scutellaris* was a junior synonym of *L. oregmae* [8]. Since this time, additional species of *Lipolexis* have been reported. *Lipolexis chinensis* Chen was described by Chen [9], but three years later Starý and Ghosh [10] placed it as a junior synonym of *L. gracilis*. However, three more species were subsequently described from the Oriental region: *Lipolexis wuyiensis* Chen [11] from China, and both *Lipolexis myzakkaiae* Pramanik and Raychaudhuri and *Lipolexis pseudoscutellaris* Pramanik and Raychaudhuri [12] from India. Although these three species are still regarded as valid, their descriptions leave doubt. Moreover, it seems that neither species has been mentioned in the scientific literature since their description. *Lipolexis gracilis* has a Palaearctic distribution and it is the only *Lipolexis* currently thought to be widely distributed in Europe [13,14,15]. By contrast, *L. oregmae*, an Oriental species, has been introduced to USA (Florida) [16], Jamaica [17], Dominica [18] and Costa Rica [19]. While the potential value of *L. gracilis* in biocontrol of black bean aphid, *Aphis fabae* Scopoli [20], and soybean aphid, *Aphis glycines* Matsumura [21] has been considered, *L. oregmae* has proven effective against the brown citrus aphid, *Toxoptera citricida* (Kirkaldy). The latter aphid causes serious damage to citrus plantations both by transmitting the tristeza closterovirus [22] and by direct damage to the plant. *Lipolexis oregmae* has been successfully introduced to Florida from Guam in a biological control program directed against this aphid pest [16]. Hoy et al. [17] reported the unintentional establishment of *L. oregmae* in Jamaica, where the invasion of *T. citricida* has led to serious losses in citrus production. Furthermore, Cocco et al. [18] and Zamora et al. [19] reported the same situation for Dominica and Costa Rica, respectively.

As Mackauer [23] considered *Lipolexis* as phylogenetically close to the genera *Trioxys* and *Binodoxys*, he placed it within the subtribe Trioxina. This subtribe is classified by different authors within tribe Trioxini [24] or tribe Aphidiini [23]. However, morphological examination of its immature stages led Finlayson [25] to recognize that larvae of *Lipolexis* possess an epistoma, a unique character for Aphidiinae genera. Based on this observation, she suggested that *Lipolexis* should be placed in a separate subtribe despite the morphological similarity of the adult to those of Trioxina species.

Although the genus *Lipolexis* is widely distributed and one species is an important biological control agent, the last taxonomic study on it was conducted more than 50 years ago by Starý [3]. Several molecular studies on Aphidiinae included representatives of this genus, but conclusions concerning the position of *Lipolexis* were ambiguous. Based on phylogenetic analysis of the mitochondrial NADH 1 dehydrogenase gene using neighbor joining analysis, Smith et al. [26] positioned *L. gracilis* as a sister group to the rest of Aphidiina and Trioxina species. In the same study, both weighted and unweighted parsimony analysis positioned *L. gracilis* within the tribe Aphidiini. Furthermore, in the weighted parsimony analysis of the 16S RNA sequence conducted by Kambhampati et al. [27], *L. gracilis* was placed as a sister group to the genera *Binodoxys* and *Trioxys* with high bootstrap value supporting placement of *Lipolexis* within the subtribe Trioxina. However, the same study placed *L. gracilis* with the subtribe Aphidiina with low bootstrap support in the unweighted parsimony analysis. The results of phylogenetic analysis of 18S RNA region [28] positioned *L. gracilis* as a sister group to the genus *Trioxys*, supporting Mackauers’ [23] classification. The study of Shi and Chen [29] based on three genes (16S RNA, 18S RNA, ATPase 6) also supports the traditional classification of *Lipolexis* within the subtribe Trioxina. Furthermore, analysis of the barcode region of cytochrome *c* oxidase subunit I (COI) yielded a phylogenetic tree with *L. gracilis* grouped as a sister taxon to *Falciconus pseudoplatani* Marshall within species in the subtribes Trioxina and Monoctonina [30]. Based on these results, it is evident that the position of *Lipolexis* within Aphidiinae remains uncertain and merits further research.

Our exhaustive studies on populations of *L. gracilis* across Europe and *L. oregmae* in the far East and USA (Florida) revealed considerable variation in taxonomically important morphological characters, such as the number of maxillary and labial palpomeres, as well as in the shape of flagellomere 1 (F1) and petiole. Motivated by these observations, both morphology and molecular analysis (COI barcode region) were used to re-examine specimens of *L. gracilis* and *L. oregmae* from various sites across their distributions. Our research sought to ascertain if *L. gracilis* and *L. oregmae* include cryptic species and to reconstruct phylogenetic relationships within the genus. It led to recognition of six new species and a key for the identification to all known *Lipolexis* species.

## 2. Materials and Methods

### 2.1. Sample Collection

Specimens were collected over 12 years at sites throughout Europe (Serbia, Montenegro, Croatia, Belgium, Slovenia, Bosnia and Herzegovina, Greece, Spain and Czech Republic), Middle East (Turkey, Iran, Pakistan), in the Far East (Bangladesh, China) and in USA (Florida) (Appendix A). Parasitoids were collected by net sweeping and Malaise traps by the Barcode of Life initiative or by sampling parts of the plant infested with aphid colonies. In the latter case, plants with aphid colonies were placed in plastic containers covered by nylon tulles to allow ventilation, and transported to the laboratory where they were held under controlled conditions (22.5 °C, 65% relative humidity, 16 h light/8 h dark). Upon emergence, parasitoids were preserved in 96% ethanol. To establish tri-trophic associations, aphid and plant samples were identified to a species or genus level. Parasitoid specimens were examined under the ZEISS Discovery V8 (Carl Zeiss MicroImaging GmbH, Göttingen, Germany) or Olympus SZX9 (Olympus, Tokio, Japan) stereomicroscopes before being dissected and slide mounted. Slides were photographed with a Leica DM LS phase contrast microscope (Leica Microsystems GmbH, Wetzlar, Germany), while relevant measurements were made using the ImageJ [31] software. Terminology of morphological characters follows Sharkey and Wharton [32]. The parasitoids examined in this study are deposited in the collection at the Institute of Zoology, Faculty of Biology, University of Belgrade (FBUB), in the Croatian Natural History Museum, Zagreb, Croatia, and in the Canadian National Collection of Insects, Arachnids, and Nematodes, Ottawa (CNC). Specimen collector abbreviations are following: ŽT (Željko Tomanović), KK (Korana Kocić), KKos (Katarina Kos), NK (Nickolas Kavallieratos), YA (Yahana Aparicio), VŽ (Vladimir Žikić), AP (Andjeljko Petrović), Jelisaveta Čkrkić (JČ), Petr Starý (PS) and MK (Marina Kaiser).

### 2.2. Molecular Analysis

Genomic DNA was extracted nondestructively with the QIAGEN Dneasy^®^ Blood & Tissue Kit (Qiagen Inc., Valencia, CA, USA) from 23 specimens, following the protocol provided by the manufacturer. Universal primers LCO1490 and HCO2891 were used for amplification of the barcode COI region [33]. The final volume of the amplification mixture was 20 µL, which contained 11.8 µL of nuclease-free water, 0.2 µL of Taq Polymerase, 1 µL of primers (each), 1.2 µL of nucleotides, 1.8 µL of MgCl_2_, 2 µL of buffer and finally 1 µL of the extracted DNA. The following PCR temperature and cycle program was used: 5 min of initial denaturation, 35 cycles of 60 s denaturation (94 °C), 60 s annealing (54 °C) and 90 s extension (72 °C) and 7 min of final extension (72 °C). Amplification products were purified with QIAquick PCR purification Kit (Qiagen Inc., Valencia, CA, USA) and sequenced by Macrogen Inc. (Seoul, Korea). After DNA extraction, specimens were examined morphologically.

### 2.3. Phylogenetic Analysis

Electropherograms were visualized in Finch TV Geospiza Inc. (Seattle, WA, USA) and manually edited and aligned using BioEdit [34] software. The analysis of evolutionary divergence was conducted in MEGA 6 [35] software using the Kimura 2-parameter distance model. The Tamura 3-parameter model [36] with gamma distribution and invariate sites (T92+G+I) was indicated by MEGA 6 as the best fitting model test. Bayesian evolutionary analysis was performed with BEAST 2.5 [37] software employing the initial data set constructed in BEAUti v1.10.4 [37] with designated strict clock type and Yule process of speciation. The analysis ran for 10 million generations, the sampling was conducted every 1000 generations, while the first million trees were discarded as a burn in. The effective sample size (ESS) of the parameters of the Markov chain Monte Carlo was estimated by Tracer v1.7.1 [38]. The saturation level for the third codon position was inspected in DAMBE software [39] using Xia model test [40]. The phylogenetic tree was visualized using FigTree 1.4.3 software [41]. Maximum likelihood (ML) and Neighbor Joining (NJ) phylogenetic trees were constructed in MEGA 6 (Tamura 3-parameter model, 2000 replicates, total of 683 positions in the final dataset). In all analyses, the outgroup species was *Venturia canescens* (Gravenhorst). Haplotype diversity was estimated by the software DNAsp [42] and haplotype networks were constructed by Network (version 5.0.0.1). The final dataset contained 64 COI sequences (one outgroup sequence, additional 40 sequences acquired from BOLD systems database (25) and GenBank (15), and 23 newly recovered sequences).

## 3. Results

### 3.1. Molecular Analysis

COI sequences were acquired from 63 specimens of *Lipolexis*, five assigned to *L. oregmae* and 58 to *L. gracilis* (Appendix A). All three phylogenetic reconstruction methods (Bayesian, ML, NJ) yielded a tree with similar topology (Figure 1). The sequences fell into two main lineages (Table 1) showing deep sequence divergences (22.7–23.2%); the first group included all 58 individuals identified as *L. gracilis* (“*gracilis*” group) while the second included the five specimens of *L. oregmae* (“*oregmae*” group) (Figure 1).

The five sequences initially assigned to *L. oregmae* were separated into two distinct clades with an average genetic distance of 19.9%. A haplotype network supported the presence of two species (*L. oregmae*, *Lipolexis bengalensis* sp. n) with high sequence divergence (77 mutational steps) at COI (Figure 2).

The 58 sequences recovered from members of *L. gracilis* group included 26 haplotypes that fell into two main groups (“group 1” and “group 2”) (Figure 1) with the average sequence divergence of 11.4%. The first group included European specimens which were separated into two clusters with an average COI distance of 7.3% (*L. gracilis* s.s., *Lipolexis labialis* sp. n.). The second group included three clades: the first represented by a single specimen from China (*Lipolexis takadai* sp. n.) with an average genetic distance of 4.5% and 7.2%, from the other two clades. The second clade (*Lipolexis pelopsi* sp. n.) included six individuals from the Mediterranean, while the third included specimens from the Middle East, the Orient, and Spain. The average evolutionary divergence between the Mediterranean and the third clade was 8%. The Middle East and Oriental region clade, which also contained several specimens from Spain, was further separated into two clades (*Lipolexis pakistanicus* sp. n. and *Lipolexis peregrinus* sp. n.) with mean distance of 2.2%. The 26 haplotypes of *L. gracilis* s.l. (Figure 3) were separated into six distinct groups that corresponded to the named clades shown in the phylogenetic tree.

Morphological analysis of the *Lipolexis* specimens in each clade revealed diagnostic differences supporting the recognition of six new species which is in concordance with the phylogenetic results. The balance of the Results section describes the new taxa and redescribes *L. gracilis* and *L. oregmae*.

### 3.2. Description of Six New Species

#### 3.2.1. *Lipolexis bengalensis* sp. n. Tomanović and Kocić

http://zoobank.org/46F9F556-F290-4BD0-B5A3-69390B4C78C0.

*Diagnosis*. *Lipolexis bengalensis* sp. n. possesses crenulated longitudinal carinae along the sides of petiole and belongs to the *oregmae* group. However, it differs from the nominative species, *L. oregmae*, because its maxillary palps have two palpomeres (*L. oregmae* has three maxillary palpomeres). Additionally, *L. bengalensis* sp. n. has a shorter F1 (proportions between length and maximum width at the middle of F1 are 4.0 in *L. bengalensis* sp. n. vs. 4.45 in *L. oregmae*) and a stouter petiole (proportions between length and width of petiole at spiracles 2.6–2.9 in *L. bengalensis* sp. n. and 3.3 in *L. oregmae*) (Figure 4).

##### Female

*Head* (Figure 4B). Head wider than mesosoma (proportion between head and mesoscutum width 1.40–1.50), with sparse long setae. Eyes large, oval, laterally prominent. Face uniformly pubescent with moderately long setae. Clypeus protrudent with 6–7 long setae. Tentorial index 0.20. Malar space 0.15× as long as longitudinal eye diameter. Mandible bidentate, with 8–10 long setae on outer surface. Maxillary palps with two palpomeres. Labial palps with one palpomere. Antenna damaged (Figure 4A) (antenna broken, missing F8–F10), filiform; antennal segments long and cylindrical with semi-erected long sparse setae that are longer than half of flagellomeres diameter. F1 (Figure 4A) equal to F2; F1 and F2 bearing 1–2 and 2–3 longitudinal placodes, respectively. Proportion between length and maximum width at middle of F1 and F2, 4.0.

*Mesosoma*. Mesoscutum (Figure 4C) smooth, covering pronotum above; notaulices distinct in very short ascedent portion of anterolateral margin, effaced dorsally, with a series of 6–7 long setae along the latero-dorsal part of mesoscutum, almost reaching to scutellum. Scutellum nearly triangular, bearing 2 long setae on lateral margins. Propodeum (Figure 4D) clearly areolated, with a wide central areola, with pronounced oblique antero-central carinae extending to the spriracles. External and dentiparal areolae of propodeum with 5–6 and 1–2 long setae in each side, respectively. Forewing (Figure 4G) densely pubescent, marginal setae longer than the surface setae; venation hyaline; pterostigma triangular, 2.3× as long as wide and equal to R1 vein; vein r and RS long, reaching near the apex of the wing.

*Metasoma*. Petiole (Figure 4E) long and slender, slightly widened at apex and narrow posteriorly; its length 2.6–2.9× as long as wide at spiracles; petiole is dorsally smooth, but it bears crenulated longitudinal carinae along the sides, a feature typical for the *oregmae* group, and two long setae near the base of petiole at each side. Ovipositor sheath long (Figure 4F), wide at base and curved downwards, bearing scattered very long setae in the middle portion; distally dilated slightly. Length of ovipositor sheath 2.5–2.8× as long as maximum width at base and 7.3–7.6× as long as minimum width at tip. Second valvulae with a convex dorsal outline, second valvifer narrow, widened at the joint in point with ovipositor sheath.

*Body length*. 1.5–1.8 mm.

*Coloration*. Head dark brown, face brown. Mouthparts yellow to light brown. Scape and pedicel light brown. F1 segment brown with very narrow yellow or light brown ring at the base, remaining antennal flagellomere brown. Mesosoma brown with light brown legs. Metasoma dark brown. Ovipositor sheath light brown.

##### Male

Unknown.

Distribution: Bangladesh.

Etymology: *L. bengalensis* sp. n. takes its name after Bengal, the old name of Bangladesh, reflecting the current known distribution of this species.

*Material*. Holotype: 1♀, 12. IV 2014, Chittagong, Bangladesh (BOLD ID: GMBCJ2865-15). Paratypes: 1♀, 12. IV 2014, Chittagong, Bangladesh (BOLD ID: GMBCJ1757-15); 2♀, 11.VI 2014, Chittagong, Bangladesh (BOLD ID: GMBCI2841-15, GMBCI3219-15). Holotype is slide mounted and deposited in the Canadian National Collection of Insects, Arachnids and Nematodes, Ottawa (CNC). Paratype specimens are slide mounted and deposited in the collection of Institute of Zoology, Faculty of Biology, University of Belgrade (FBUB).

#### 3.2.2. *Lipolexis labialis* sp. n. Tomanović and Kocić

http://zoobank.org/F19883FA-0504-43B0-8D60-4B8ECAE90327.

*Diagnosis*. *L. labialis* sp. n. possesses maxillary palps with four palpomeres, a feature shared with two other members of the *gracilis* group—*L. gracilis* and *L. pelopsi* sp. n. However, it is the only known species of *Lipolexis* with labial palps with two palpomeres. Additionally, it differs from *L. gracilis* and *L. pelopsi* sp. n. by having a more elongated F1 (proportions between length and maximum width at middle of F1, 3.7–4.0 in *L. labialis* sp. n. vs. 3.0–3.6 in *L. gracilis* and 3.4–3.6 in *L. pelopsi* sp. n.) and generally smaller number of longitudinal placodes on F1 and F2 than *L. gracilis* and *L. pelopsi* sp. n. (*L. labialis* sp. n. has 1–2 on F1 and 2–4 longitudinal placodes on F2, while *L. gracilis* has 2–4 on F1 and 4–6 on F2 and *L. pelopsi* sp. n. has 3–4 on F1 and 4–5 on F2) (Figure 5).

##### Female

*Head* (Figure 5B) transverse, wider than mesosoma at tegulae (proportion between wide of head and mesoscutum 1.37–1.45), with sparse long setae. Eyes large, oval, laterally prominent. Clypeus with 6–7 long setae. Tentorial index 0.28–0.31. Malar space equal to 0.18–0.20× as long as longitudinal eye diameter. Mandible bidentate, with 7–10 long setae on outer surface. Maxillary palps with four palpomeres. Labial palps with two palpomeres. Antenna 12—segmented (Figure 5A), filiform; antennal segments long and cylindrical with semi-erected long sparse setae equal to about half of flagellomeres diameter or little longer. F1 (Figure 5A) equal or slightly longer than F2; F1 and F2 bearing 1–2 and 2–4 longitudinal placodes, respectively. Proportions between length and maximum width at middle of F1 and F2, 3.7–4.0 and 3.3–4.0, respectively.

*Mesosoma*. Mesoscutum (Figure 5C) smooth, covering pronotum above; notaulices distinct in very short ascendant portion of anterolateral margin, effaced dorsally, with a series of 8–10 long setae along the latero-dorsal part of mesoscutum. Scutellum nearly triangular, bearing 2–3 long setae on lateral margins. Propodeum (Figure 5D) areolated, with a wide central areola, sometimes with no clear anterior carinae. External and dentiparal areolae of propodeum with 2–3 and 1 long setae in each side, respectively. Central areola with rugosities. Forewing (Figure 5G) densely pubescent, marginal setae longer than the surface setae; pterostigma triangular, 2.5–2.9× as long as wide and subequal to R1 vein (proportion between pterostigma length and R1 vein length 0.9–1.1); vein r and RS long, reaching near the apex of the wing.

*Metasoma*. Petiole (Figure 5E) long and slender, slightly widened at apex; its length 2.7–3.2× as long as wide at the weak prominent spiracles; a pair of central carinae, distinctly prominent along dorsal surface of the petiole, diverging toward base. Ovipositor sheath strong, long and slightly curved (Figure 5F), dilated at the apex. Length of ovipositor sheath 2.5× as long as wide at the base, and approx. 10× as long as tip.

*Body length*: 1.6–1.8 mm.

*Coloration*: Head brown, face brown. Mouthparts yellow. Scape, pedicel and F1 yellow till light brown, remaining flagellomeres brown. Mesosoma brown. Metasoma brown with light brown petiole and ovipositor sheath.

##### Male

Antennae 13—segmented. F1 and F2 shorter than in females, 2.8–3.1× as long as wide and 3.0–3.1× as long as wide, respectively. Pterostigma 2.44–2.75× as long as wide. Aedeagus funnel shaped, apodemae long, volsellae prominent and pointed; paramerae short, triangular, bearing 4–5 long setae (Figure 5H). Color same as in female.

*Body length*: 1.4–1.6 mm.

*Hosts*: *Anoecia corni* (F.), *Dysaphis* sp. Börner, *Myzus cerasi* (F.), *Myzus lythri* (Schrank), *Roepkea marchali* (Börner).

*Distribution*. European distribution. Recorded from Serbia, Croatia, Spain, Montenegro, Slovenia, and Bulgaria.

*Etymology*. *Lipolexis labialis* sp. n. takes its name from its possession of the highest number of labial palpomeres (2) within *Lipolexis*.

Material: Holotype: 1♀ reared from *Dysaphis* sp. Börner on *Malus* sp., 12. VI 2013, Umčari, Kotlova, Serbia; Paratypes: 1♀1♂, *Dysaphis* sp. Börner on *Malus* sp., 12. VI 2013, Umčari, Kotlova, Serbia (MK); 3♀5♂, *M. cerasi* on *Prunus cerasus*, 16. VI 2013, Umčari, Parloge, Serbia (MK); 1♀1♂, *M. lythri* on *Lythrum salicaria*, 21. VI 2015, Plitvička jezera, Kozjak, Croatia (ŽT); 6♀5♂, *A. corni* on *Cornus mas*, 21.VI 2015, Plitvička jezera, Kozjak, Croatia (ŽT); 1♀, 2016, Segria, Spain (YA); 6♀6♂, *R. marchali* (Börner) on *Prunus mahaleb,* 18. VI 2018, Lovćen, Njeguši, Montenegro (KK); 1♀2♂, *M. cerasi* (F.) on *Prunus cerasi*, 30.V 2016, Škocjanske jame, Slovenia (KK).

Holotype is slide mounted and deposited in the collection of Institute of Zoology, Faculty of Biology, University of Belgrade (FBUB). Four female and one male paratypes are slide mounted and deposited in the collection of Institute of Zoology, Faculty of Biology, University of Belgrade (FBUB). Paratypes collected in National Park Plitvice, Croatia are deposited in the Croatian Natural History Museum, Zagreb, Croatia. Two female and two male paratypes (Umčari, Parloge, Serbia) are deposited in the Canadian National Collection of Insects, Arachnids and Nematodes, Ottawa (CNC).

#### 3.2.3. *Lipolexis takadai* sp. n. Tomanović and Kocić

http://zoobank.org/95FFBE0B-087E-41D0-A2A4-0D61D0938D48.

*Diagnosis*. *Lipolexis takadai* sp. n. differs from all known species of *Lipolexis* by the following combination of morphological characters: maxillary palps with three palpomeres and labial palps with one palpomere, and very long F1 and F2 (proportions between length and maximum width at middle of F1 and F2, 4.75 and 4.80–5.0, respectively). All known *Lipolexis* species possess proportions between length and maximum width at middle of F1 and F2, 3.0–4.45 and 2.70–4.25, respectively (Figure 6).

##### Female

*Head* (Figure 6B) is wider than mesosoma at tegulae (proportion between head and mesoscutum width 1.30), with sparse long setae. Eyes large, oval, laterally prominent. Face uniformly pubescent with moderately long setae. Clypeus protrudent with 7–8 long setae. Tentorial index 0.30–0.35. Malar space equal to 0.15–0.20× as long as longitudinal eye diameter. Mandible bidentate, with 8–10 long setae on outer surface. Maxillary palps with three palpomeres. Third palpomere long but undivided. Labial palps with one palpomere. Antenna 12—segmented, filiform (Figure 6A); antennal segments long and cylindrical with semi-erected long sparse setae length about half of flagellomeres diameter. F1 (Figure 6A) longer than F2; F1 and F2 bearing 2 and 3 longitudinal placodes, respectively. Proportions between length and maximum width at the middle of F1 and F2, 4.75 and 4.8–5.0, respectively.

*Mesosoma*. Mesoscutum (Figure 6C) smooth, covering pronotum above; notaulices distinct in very short ascedent portion of anterolateral margin, effaced dorsally, with a series of 9–10 long setae along the latero-dorsal part of mesoscutum, reaching to scutellum. Scutellum nearly triangular, noticeably crenulated along lateral margins, bearing 2–3 long setae. Propodeum (Figure 6D) areolated, with a wide central areola, oblique antero-central carinae extending to the spriracles. External and dentiparal areolae of propodeum with 3–4 and without long setae in each side, respectively. Forewing (Figure 6G) densely pubescent, marginal setae longer than the surface setae; venation hyaline; pterostigma triangular, 2.6–2.7× as long as wide and subequal to R1 vein; vein r and RS long, reaching near to the apex of the wing.

*Metasoma*. Petiole (Figure 6E) long and slender, slightly widened at apex; its length 2.8× as long as wide at spiracles, with prominent central carina. Ovipositor sheath (Figure 6G) long and slender, strongly curved downwards, wide at base, narrowing in the last third and slightly wider at the tip. Length to width ratio of the ovipositor sheath is 2.7 at the widest site, and 10 at the tip.

*Body length*: 1.5–1.8 mm.

*Coloration*: Head brown. Mouthparts yellow. Scape and pedicel yellow, F1 light brown, remaining antennal segments brown. Mesosoma brown. Metasoma brown with light brown petiole. Ovipositor sheath light brown.

##### Male—Unknown

*Distribution*. Japan and China.

*Host*. *Aphis gossypii* Glover.

*Etymology*. *Lipolexis takadai* sp. n. is named in honor of the famous Japanese entomologist and taxonomist, Prof. Hajimu Takada, for his exceptional contribution to the knowledge of the parasitoid fauna of aphids in Japan, Far East and worldwide.

*Material*. Holotype: 1♀, 27. VII 2012, Shaanxi, Yinggezhen, China (BOLD ID: GMHCN 462-14); Paratypes: 1♀, reared from *Aphis gossypii* Glover on *Solanum melongena*, 25. VII 1962, “Moraica”, Japan; 1♀, reared from *A. gossypii* Glover, Kyoto, Japan, leg. Hajimu Takada. Holotype is slide mounted and deposited in the Canadian National Collection of Insects, Arachnids and Nematodes, Ottawa (CNC). Paratypes are slide mounted and deposited in the collection of Institute of Zoology, Faculty of Biology, University of Belgrade (FBUB).

#### 3.2.4. *Lipolexis pelopsi* sp. n. Tomanović and Kavallieratos

http://zoobank.org/DB86D778-3771-4DD3-AF97-9A5E57165B1A.

*Diagnosis*. *Lipolexis pelopsi* sp. n. differs from all known *Lipolexis* species by its combination of the following morphological characters: maxillary palps with four palpomeres, labial palps with one palpomere (the same combination as *L. gracilis*), pubescent body and stout petiole (proportions between length and maximum width of petiole at spiracles, 2.4–2.6 in *L. pelopsi* sp. n., vs. 3.0–3.6 in *L. gracilis*) (Figure 7).

##### Female

*Head* (Figure 7B) transverse, wider than mesosoma at tegulae (proportion between wide of head and mesoscutum, 1.2–1.5), with sparse long setae. Eyes large, oval, laterally prominent. Face densely pubescent with moderately long setae. Clypeus protrudent with 9–12 long setae. Tentorial index 0.25–0.35. Malar space equal to 0.12–0.18× as long as longitudinal eye diameter. Maxillary palps with four palpomeres. Labial palps with one palpomere. Antenna 12—segmented (Figure 7A), filiform; antennal segments long and cylindrical with semi-erected long sparse setae shorter than half of flagellomeres diameter. F1 (Figure 7A) subequal to F2; F1 and F2 bearing 3–4 and 4–5 longitudinal placodes, respectively. Proportions between length and maximum width at middle of F1 and F2, 3.4–3.6 and 3.6, respectively.

*Mesosoma*. Mesoscutum (Figure 7C) smooth, covering pronotum above; notaulices distinct in very short ascedent portion of anterolateral margin, effaced dorsally, with a series of 6 to 10 long setae along the latero-dorsal parts of mesoscutum, reaching to scutellum. Specimens collected in Greece possess densely pubescent mesoscutum with 18 to 20 setae. Scutellum nearly triangular, bearing 5–6 setae on lateral margins. Propodeum (Figure 7D) areolated, with a wide central areola, sometimes with undefined upper carinae. External and dentiparal areolae of propodeum with 5–6 and 0–1 long setae in each side, respectively. Forewing (Figure 7G) densely pubescent, marginal setae long and longer than the surface setae; venation reduced; pterostigma triangular, 2.3–2.6× as long as wide and subequal to R1 vein; vein r and RS long, reaching near to the apex of the wing.

*Metasoma*. Petiole (Figure 7E) long and slender, slightly widened at apex; its length 2.4–2.6× as long as wide at spiracles, slightly prominent in lateral margin; a pair of central carinae short, prominent along dorsal surface of the petiole; Ovipositor sheath narrow and curved downwards (Figure 7F), dilated at the apex, bearing 3 large long setae in the middle portion. Ovipositor 3.1× as long as wide at the widest part and 9.1× as long as wide at the tip.

*Body length*: 1.5–2.0 mm.

*Coloration*: Head brown, eyes black; mouthparts yellow. Scape, pedicel and F1 light brown, remainder of antennae brown. Mesosoma brown. Legs light brown. Petiole yellow. Metasoma dorsally dark brown.

##### Male

Antennae 13–segmented, F1 and F2 bearing 3 and 4 longitudinal placodes, respectively. Pterostigma 2.70× as long as wide and clearly shorter than R1 vein. Aedeagus long, funnel shaped; volsellae strong, apodemae long (Figure 7H). Paramerae short, pointed apically, bearing 3–4 long setae.

*Body length*: 1.4–1.6 mm.

*Distribution*. Mediterranean distribution, recorded from Bosnia and Herzegovina, Montenegro and Greece.

*Host*. Species in the genus *Aphis* L.

*Etymology*. *Lipolexis pelopsi* sp. n. takes its name after Pelops, mythological king of Peloponnese region, which is the area of the first collected specimens.

*Material*: Holotype: 1♀ reared from *Aphis punicae* Passerini on *Punica granatum*, 19. V 2010, Tivat, Montenegro (VŽ); Paratypes: 4♀7♂, reared from *A. fabae* Scopoli on *Diervilla florida*, 23. V 2017, Blagaj, Bosnia and Herzegovina (ŽT); 1♂, reared from *Aphis hederae* Kaltenbach on *Hedera helix*, 23. V 2017, Buna, Bosnia and Herzegovina (ŽT); 1♀ reared from *Aphis* sp. on *Citrus deliciosa*, 18. V 2010, Petrovac, Montenegro (VŽ); 4♀4♂, reared from *Aphis ruborum* (Börner and Schilder) on *Rubus* sp., 15. VI 2018, Rijeka Crnojevića, Montenegro; 8♀4♂, reared from *Aphis fabae cirsiiacanthoides* Scopoli on *Cirsium arvense*, 01. V 2010, Kyparissia, Greece (ŽT, NK). Holotype and 5♀1♂ paratypes slide mounted and deposited in the collection of Institute of Zoology, Faculty of Biology, University of Belgrade (FBUB) (the rest of specimens are kept in 96% ethanol). Two female and two male paratypes (Blagaj, Bosnia and Herzegovina) are deposited in the Canadian National Collection of Insects, Arachnids and Nematodes, Ottawa (CNC).

#### 3.2.5. *Lipolexis pakistanicus* sp. n. Tomanović and Kocić

http://zoobank.org/54B5715D-2105-42A2-9AD4-17F53319243E.

Diagnosis. *Lipolexis pakistanicus* sp. n. belongs to “*gracilis*” group and shares the same number of maxillary (three) and labial palpomeres (one) as *L. peregrinus* sp. n. and *L. takadai* sp. n. However, it has a more elongate pterostigma than these species (2.9 length to width ratio in *L. pakistanicus* sp. n., vs. 2.6–2.7 and 2.4–2.7 in *L. takadai* sp.n. and *L. peregrinus* sp. n., respectively) (Figure 8).

##### Female

*Head* (Figure 8B) wider than mesosoma (proportion between head width and mesoscutum width 1.5), with sparse long setae. Eyes large, oval, laterally prominent. Face uniformly pubescent with moderately long setae. Clypeus protrudent with 6–7 long setae. Tentorial index 0.30. Malar space equal to 0.20× as long as longitudinal eye diameter. Mandible bidentate, maxillary palps with three palpomeres and labial palps with one palpomere. Antenna 12—segmented (Figure 8A), filiform; antennal segments long and cylindrical with semi-erected long sparse setae length about half of flagellomeres diameter. F1 (Figure 8A) equal or subequal to F2; F1 and F2 bearing 3–4 and 4–5 longitudinal placodes, respectively. Proportion between length and maximum width at middle of F1 and F2, 4.25.

*Mesosoma*. Mesoscutum (Figure 8C) smooth, covering pronotum above; notaulices distinct in very short ascedent portion of anterolateral margin, effaced dorsally, with a series of 6–7 long setae along the latero-dorsal part of mesoscutum, reaching to scutellum. Scutellum nearly triangular, slightly crenulated along lateral margins, bearing 2–3 long setae on lateral margins. Propodeum (Figure 8D) clearly areolated, with a wide central areola, oblique antero-central carinae extending to the spriracles. External and dentiparal areolae of propodeum with 3–4 and without long setae in each side, respectively. Forewing (Figure 8G) densely pubescent, marginal setae longer than the surface setae; venation reduced; pterostigma triangular, 2.9× as long as wide and little shorter than R1 vein (proportion between pterostigma length and R1 vein length 0.90); vein r and RS long, reaching near to the apex of the wing.

*Metasoma*. Petiole (Figure 8E) slightly widened at apex; its length 2.7× as long as wide at spiracles, prominent in lateral margin; dorsally smooth but it bears short and strong mediodorsal carina with two short longitudinal carinae diverging along the sides to the posterior part of petiole. Petiole bears two very long setae along both sides near base. Ovipositor sheath (Figure 8F) long, wide at base, curved downwards, with no visible setae; distally more dilated than in other species, upper part of the ovipositor sheath more sclerotized along the whole length. Length of ovipositor sheath 2.56× as long as wide at base and 6.5× as long as minimum width at tip.

*Body length*: 1.5–1.8 mm.

*Coloration*: Head dark brown, face brown. Mouthparts yellow to light brown. Scape, pedicel and base of F1 yellow to light brown, remaining parts of antenna brown. Mesosoma brown with light brown legs. Metasoma brown or light brown with yellow petiole.

##### Male

F1 subequal to F2. Mesoscutum smooth, with a series of 7–8 long setae. Propodeum with clear central areola. Petiole shorter and broader than in female. Aedeagus funnel shaped, pointed apically, volsellae strong and triangular (Figure 8H). Parameare short, bearing 3–4 setae. Head testaceus, antennae, mesoscutum, propodeum and abdomen brown, legs yellow.

*Host*. *Aphis gossypii* Glover.

*Distribution*. Pakistan, Bangladesh and Moldova.

*Etymology*. *Lipolexis pakistanicus* sp. n. takes its name from the country where the majority of specimens were collected.

*Material*: Holotype: 1♀, 28. III 2012, Islamabad, Pakistan (BOLD ID: MAMTG667-12); Paratypes: 1♀, 11. IV 2012, Islamabad, Pakistan (BOLD ID: MAMTI575-12); 1♀1♂, 15. XI 2012, Islamabad, Pakistan (BOLD IDs: MAMTW303-14, MAMTW306-14); 1♀, 12. IV 2014, Chittagong, Bangladesh (BOLD ID: GMBCA4763-15), 1♀, reared from *A. gossypii* on *Cucumis* sp., 26. VII 1972, Pakistan; 2♀, Moldova, date and location record of these samples is unknown.

The holotype is slide mounted and deposited in the Canadian National Collection of Insects, Arachnids and Nematodes, Ottawa (CNC). Paratypes are slide mounted and deposited in the collection of Institute of Zoology, Faculty of Biology, University of Belgrade (FBUB).

#### 3.2.6. *Lipolexis peregrinus* sp. n. Tomanović and Kocić

http://zoobank.org/F82ED4DF-913A-4242-9005-8CBADEEB7CD9.

*Diagnosis*. *Lipolexis peregrinus* sp. n. belongs to “*gracilis*” group and possesses three maxillar palpomeres and one labial palpomere as *L. takadai* sp. n. and *L. pakistanicus* sp. n. However, it clearly differs from both species by having shorter F1 (proportions between length and maximum width at middle of F1 in *L. peregrinus* sp. n., 3.4–3.8 vs. 4.75 in *L. takadai* sp. n. and 4.25 in *L. pakistanicus* sp. n., respectively) and more elongated petiole (proportion between length and width of petiole at spiracles in *L. peregrinus* sp. n., 3.1–3.3 vs. 2.8 in *L. takadai* sp. n. and 2.7 in *L. pakistanicus* sp. n., respectively (Figure 9).

##### Female

*Head* (Figure 9B) transverse, wider than mesosoma at tegulae (proportion between width of head and mesoscutum 1.40–1.50), with sparse long setae. Eyes large, oval, laterally prominent. Face with moderately long setae. Clypeus with 7–8 long setae. Tentorial index 0.23–0.30. Malar space equal to 0.13–0.18× as long as longitudinal eye diameter. Mandible bidentate, with 11–12 long setae on outer surface. Maxillary palps with three palpomeres. Labial palps with one palpomere. Antenna 12—segmented (Figure 9A), filiform; antennal segments long and cylindrical with semi-erected long setae length about half of flagellomeres diameter. F1 (Figure 9A) equal to F2 or somewhat longer; F1 and F2 bearing 2–3 and 4–5 longitudinal placodes, respectively. Proportions between length and maximum width at middle of F1 and F2, 3.4–3.8 and 3.2–3.6, respectively.

*Mesosoma*. Mesoscutum (Figure 9C) smooth, covering pronotum above; notaulices distinct in very short ascedent portion of anterolateral margin, effaced dorsally. Mesoscutum sparsely setous with a 5–7 long setae along the latero-dorsal parts. Scutellum nearly triangular, bearing 1–3 long setae on lateral margins. Propodeum (Figure 9D) areolated, with a wide central areola, oblique antero-central carinae clearly extending to the spriracles. External and dentiparal areolae of propodeum with 3–5 and 0–1 long setae on each side, respectively. Forewing (Figure 9G) densely pubescent with setae longer than the ones on surface; pterostigma triangular, 2.4–2.7× as long as wide and shorter than R1 vein (proportion between pterostigma length and R1 vein length 0.90); vein r and RS long, reaching near the apex of the wing.

*Metasoma*. Petiole (Figure 9E) long and slender, slightly widened at apex; length 3.1–3.3× as long as wide at spiracles, slightly prominent at lateral margin; a pair of central carinae, distinctly prominent along dorsal surface of the petiole, merged or separate in anterior part, but diverging toward base; one long seta at the base of each lateral side; spiracular tubercules smooth, positioned beyond the first half of the segment. Ovipositor sheath (Figure 9F) long, wide at base, curved downwards; distally dilated slightly, upper part of the ovipositor sheath more sclerotized along the whole length. Length of ovipositor sheath 2.8–2.9× as long as maximum width at base and 8.0–9.0× as long as minimum width at tip.

*Body length*: 1.5–2.0 mm.

*Coloration*: Head light brown. Mouthparts yellow. Scape, pedicel and F1 yellow to light brown, remaining antennal segments brown. Mesosoma and metasoma light brown to brown. Petiole light brown.

##### Male

Antennae 13—segmented. Flagellomeres somewhat shorter than in females. F1 and F2, 3.30 and 3.60× as long as wide, respectively. Pterostigma more triangular than in female (2.20–2.30× as long as wide). Aedeagus funnel shaped, apodemae long; volsellae wide, triangular. Paramerae short, bearing 3 setae (Figure 9H). Body darker than in females with yellow legs and mouthparts.

*Body length*: 1.4–1.6 mm.

*Hosts*. *Myzus persicae* (Sulzer), *Aphis* sp., *A. gossypii* Glover, *Toxoptera aurantii* (Boyer de Fonscolomber), *Melanaphis sacchari* (Zehntner).

*Distribution*. Europe (Spain and Slovenia) and Oriental region (China and Japan—mined from GenBank, see Appendix A for sample information).

*Etymology*. *Lipolexis peregrinus* sp. n. takes its name in regard to its unknown place of origin (latin *peregrinus* = foreign, from abroad).

*Material*. Holotype: 1♀ reared from *M. persicae* on *P. persica*, 04. VI 2015, Alfarras, Spain. Paratypes: 1♀1♂, reared from *M. persicae* on *P. persica*, 04. VI 2015, Alfarras, Spain; 1♀ reared from *M. persicae* on *P. persica*, 2015, Alfarras, Lleida, Spain; 2♀ reared from *Aphis* sp. on *Solanum tuberosum*, 27. VI 2009, Zaleb, Slovenia (KKos). Holotype and paratype specimens are dissected and slide mounted, and kept in the collection of Institute of Zoology, Faculty of Biology, University of Belgrade (FBUB) Two female paratypes (Alfarras, Lleida, Spain) are deposited in the Canadian National Collection of Insects, Arachnids and Nematodes, Ottawa (CNC).

### 3.3. Re-Description of L. gracilis and L. oregmae

#### 3.3.1. *Lipolexis gracilis* Förster, 1862, re-description

*Lipolexis gracilis* Förster, 1862, Verh. Nat. Ver. Preuss.Rheinl.19:249.

*Gynocryptus pieltaini* Quilis, 1930, Eos, Madrid 7: 27–29.

*Aphidius palpator* Gautier and Bonnamour, 1931, Bull. Soc. Ent. Fr., 166–167.

*Lipolexis chinensis* Chen, 1980, Entomotaxonomia 2: 169–172.

##### Female

*Diagnosis*. *Lipolexis gracilis* has a petiole with central bifurcating carinae, a feature that positions it within the *gracilis* species group. It possesses maxillary palps with four palpomeres, labial palps with one palpomere (the same combination as *L. pelopsi*). However, it differs from the latter by a less pubescent body and elongated petiole (proportions between length and maximum width of petiole at spiracles, 3.0–3.6 in *L. gracilis* vs. 2.4–2.6 in *L. pelopsi* sp. n.).

*Head* (Figure 10B) transverse, wider than mesosoma at tegulae, with sparse long setae. Eyes large, oval, laterally prominent. Face uniformly pubescent with moderately long setae. Clypeus protrudent with 7–9 long setae. Tentorial index 0.25–0.40. Malar space equal to 0.10–0.15× as long as longitudinal eye diameter. Mandible bidentate, with 6–9 long setae on outer surface. Maxillary palps with four palpomeres. Labial palps with one palpomere. Antenna 12—segmented (Figure 10A), filiform; antennal segments long and cylindrical with semi-erected long sparse setae of length about half of flagellomeres diameter. F1 (Figure 10A) equal or slightly longer than F2; F1 and F2 bearing 2–4 and 4–6 longitudinal placodes, respectively. Proportions between length and maximum width at middle of F1 and F2, 3.0–3.6 and 2.7–3.2, respectively.

*Mesosoma*. Mesoscutum (Figure 10C) smooth, covering pronotum above; notaulices distinct in very short ascedent portion of anterolateral margin, effaced dorsally, with a series of 6–8 long setae along the latero-dorsal part of mesoscutum, reaching to scutellum. Scutellum nearly triangular, bearing 2–3 long setae on lateral margins. Propodeum (Figure 10D) areolated, with a wide central areola, oblique antero-central carinae extending to the spriracles. External and dentiparal areolae of propodeum with 3–4 and 0–1 long setae in each side, respectively. Forewing (Figure 10G) densely pubescent, the upper marginal setae slightly longer than the surface setae; venation reduced; pterostigma triangular, 2.3–2.5× as long as wide and equal or subequal to R1 vein (proportion between pterostigma length and R1 vein length 1.0–1.1); vein r and RS long, reaching near to the apex of the wing.

*Metasoma*. Petiole (Figure 10E) long and slender, slightly widened at apex; length 2.7–3.2× as long as wide at spiracles, slightly prominent in lateral margin; a pair of central carinae, distinctly prominent along dorsal surface of petiole, merged or separate in anterior part, but diverging toward base; spiracular tubercules smooth, positioned beyond the first half of the segment. Ovipositor sheath (Figure 10F) long, wide at base, distally dilated slightly and slightly curved downwards. As in all *Lipolexis* species, the upper part of the ovipositor sheath more sclerotized along the whole length. Length of ovipositor sheath 2.9× as long as maximum width at base and 9.6× as long as minimum width at tip. *Body length*: 1.5–1.8 mm.

*Coloration*: Head dark brown, face brown. Mouthparts except tips of mandibles yellow. Scape, pedicel and F1 brown, other segments darker. Pronotum brown; mesoscutum and mesopleuron dark brown. Propodeum brown. Legs light brown, hind femur and tibia slightly darker at the middle. Wings slightly infumated, venation yellowish brown. Petiole brown. Metasoma dorsally dark brown. Ovipositor sheath yellow, dorsally darker.

##### Male

Antennae 13—segmented. Aedeagus funnel-shaped, with sub-parallel lateral margin in upper part and sharply widening toward the base (Figure 10H). Apodemae distinctly longer than trunk of the aedeagus. Volsellae wide, triangular; parameare bearing two long setae at the tip. Tentorial index 0.40, malar space equal to quarter to eye length.

*Body length*: 1.4–1.6 mm.

*Hosts*. *Acyrthosiphon* Mordvilko, *Aphis* L., *Brachycaudus* van der Goot, Capitophorus van der Goot, Liosomaphis Walker, Lipaphis Mordvilko, Melanaphis van der Goot, *Metopeurum* Mordvilko, *Myzocallis* Passerini, *Myzus*, *Rhopalosiphum*, *Semiaphis* van der Goot, *Therioaphis* Walker and *Toxoptera* [43].

Material: 41♀, 12♂ reared from *Aphis* sp. on *Rumex* sp.: 13. V 2016, Novi Beograd, Serbia (KK); 22♀18♂ reared from *Aphis* sp. on *Galium aparine*: 25. V 2016, Surčinsko jezero, Serbia (KK); 1♀ reared from *Aphis schilderi* (Börner) on *Peucedanum schottii*: 27. VII 2012, Durmitor, Montenegro (AP); 2♀2♂ reared from *Aphis* sp. on *Carduus* sp.: 15. VI 2018, Cetinje, Lipska pećina, Montenegro (AP); 1♀ reared from *A. fabae* on *Rumex* sp.: 17. VI 2018, Lovćen, Montenegro (JČ); 1♀ reared from *Aphis* sp. on *Rubus* sp.: 17. VI 2018, Rijeka Crnojevića, Montenegro (JČ); 1♀ reared from *A. fabae* on *Foeniculum vulgare*: 17. VI 2018, Lovćen, Montenegro (KK); 2♀ reared from *Aphis sambuci* on *Sambucus nigra*: VI 2013, Češke Budejovice, Czech Republic (PS).

#### 3.3.2. *Lipolexis oregmae* (Gahan 1932)

*Diaeretus oregmae* (Gahan, 1932), Ann Entomol Soc Am, 25, 736–757.

*Lipolexis scutellaris* Mackauer, 1962, Entomophaga, 7, 1, 37–45.

Remarks about *L. wuyiensis*, *L. pseudosctullearis* and *L. myzakkaiae.* On the basis of its description, *L. wuyiensis* is very similar to *L. oregmae* and it parasitizes *Ceratovacuna lanigera* Zehntner (=*Oregma lanigera*), the same aphid host as *L. oregmae* in the Philippines. Furthermore, apart from the drawing of the petiole of *L. myzakkaiae*, which does not follow the original description (i.e., two lateral longitudinal wavy branched carinae), both *L. myzakkaiae* and *L. pseudoscutellaris* are morphologically similar to *L. oregmae*. Moreover, it seems that neither species has been mentioned in the scientific literature since their description. In our opinion, all three species are likely to be conspecific with *L. oregmae*. However, due the absence of type material, the authors refrained from official synonymization.

##### Female

*Diagnosis*. *Lipolexis oregmae* possesses a petiole that bears lateral longitudinal carinae, which is a morphological character of the *oregmae* group of species. It differs from the other member of the group, *L. bengalensis* sp. n. by the number of maxillary palpomeres (three maxillar palpomeres in *L. oregmae vs*, two in *L. bengalensis* sp. n.).

*Head* (Figure 11B) wider than mesosoma at tegulae (proportion between head and mesoscutum width 1.45), with sparse long setae. Eyes large, oval, laterally prominent. Face uniformly pubescent with moderately long setae. Clypeus protrudent with 6–7 long setae. Tentorial index 0.25. Malar space 0.15x as long as longitudinal eye diameter. Maxillary palps with three palpomeres and labial palps with one palpomere. Antenna 12—segmented (Figure 11A), filiform; antennal segments long and cylindrical with semi-erected long sparse setae with length of about half of flagellomeres diameter. F1 (Figure 11A) longer than F2; F1 and F2 bearing 2–3 and 3–4 longitudinal placodes, respectively. Proportion between length and width maximum width at middle of F1 and F2, 4.45×.

*Mesosoma*. Mesoscutum (Figure 11C) smooth, covering pronotum above; notaulices distinct in very short ascedent portion of anterolateral margin, effaced dorsally, with a series of 9–10 long setae along the latero-dorsal part of mesoscutum, reaching to scutellum. Scutellum nearly triangular, bearing 2 long setae on lateral margins. Propodeum (Figure 11D) clearly areolated, with a wide central areola, oblique antero-central carinae extending to the spiracles. External and dentiparal areolae of propodeum with 6–7 and with one long setae on each side, respectively. Forewing (Figure 11G) densely pubescent, marginal setae longer than the surface setae; venation reduced; pterostigma triangular, 2.2× as long as wide and equal to R1 vein; vein r and RS long, reaching near the apex of the wing.

*Metasoma*. Petiole (Figure 11E) long and slender, wide at base, narrowing towards the apex and then slightly widened; length 3.3× as long as wide at spiracles, dorsally smooth, bearing lateral longitudinal carinae. Ovipositor sheath (Figure 11F) long, wide at base, curved downwards, bearing 3 scattered long setae in the middle portion; distally dilated slightly, upper part of the ovipositor sheath more sclerotized along the whole length. Length of ovipositor sheath 2.8× as long as maximum width at base and 10× as long as minimum width at tip. *Body length*: 1.5–1.8 mm.

*Coloration*: Head dark brown, mouthparts yellow. Scape, pedicel and F1 yellow, remainder antenna brown. Mesosoma brown with light brown legs. Metasoma brown.

##### Male

Antennae 13—segmented. Aedeagus funnel-shaped, with sub-parallel lateral margin in upper part and sharply widening toward the base. Apodemae distinctly longer than trunk of the aedeagus. Tentorial index 0.40, malar space equal to quarter to eye length.

*Body length*: 1.4–1.6 mm.

*Hosts*. *Aphis*, *Cavariella* del Guercio, *Ceratovacuna* Zehtner, *Greenidea* Schouteden, *Liosomaphis*, *Myzus*, *Pentalonia* Coquerel, *Rhopalosiphum*, *Semiaphis*, *Sitobion* Mordvilko, *Toxoptera*, *Tuberolachnus* Mordvilko [43].

Material: 1♀, 26. II 2015, Chittagong, Bangladesh (BOLD ID: GMBCM 2962-15), 61♀13♂, reared from *T. citricida* (Kirkaldy) on *Citrus* sp., Florida, USA; 1♀1♂, India (date and location record of these samples is unknown).

### 3.4. Identification Key to the World Species of Genus Lipolexis


1.Petiole dorsally smooth, bearing crenulated lateral longitudinal carinae (Figure 4E and Figure 11E). ………2*oregmae* group-Petiole with prominent bifurcating central carina dorsally, without crenulated lateral longitudinal carinae (Figure 5E, Figure 6E, Figure 7E, Figure 8E, Figure 9E, and Figure 10E).…......3*gracilis* group2.Maxillary palps with 3 palpomeres, labial palps with 1 palpomere. F1 4.45× as long as wide, number of longitudinal placodes on F1 and F2, 2–3 and 4–5, respectively (Figure 11A).
*Lipolexis oregmae*
-Maxillary palps with 2 palpomeres, labial palps with 1 palpomere. F1 4× as long as wide, F2 with 2–3 number of longitudinal placodes on F1 and F2, 1–2 and 2–3, respectively (Figure 4A).*Lipolexis bengalensis* sp. n.3.Maxillary palps with 4 palpomeres.………4-Maxillary palps with 3 palpomeres.………64.Labial palps with 2 palpomeres, F1 3.7–4.0× as long as wide, number of longitudinal placodes on F1 and F2, 1–2 and 2–4, respectively (Figure 5A).*Lipolexis labialis* sp. n.-Labial palps with one palpomere, F1 3.0–3.6× as long as wide, number of longitudinal placodes on F1 and F2, 2–4 and 4–6, respectively (Figure 7A and Figure 10A).………55.Petiole 2.4–2.6× as long as wide (Figure 7E). R1 slightly longer than pterostigma (Figure 7G).*Lipolexis pelopsi* sp. n.-Petiole 2.7–3.2× as long as wide (Figure 10E). R1 equal or slightly shorter than pterostigma (Figure 10G).
*Lipolexis gracilis*
6.F1 4.2–4.7× as long as wide (Figure 6A and Figure 8A), petiole 2.7–2.8× as long as wide (Figure 6E and Figure 8E).………..7-F1 3.4–3.8× as long as wide (Figure 9A), petiole 3.1–3.3× as long as wide (Figure 9E)*Lipolexis peregrinus* sp. n.7.F1 and F2 4.7 and 4.8–5.0× as long as wide, respectively (Figure 6A), scutellum laterally crenulated along sides (Figure 6C), number of longitudinal placodes on F1 and F2 2 and 3, respectively (Figure 6A)*Lipolexis takadai* sp. n.-Both F1 and F2 4.2× as long as wide (Figure 8A), scutellum laterally not bearing crenulations (Figure 8C), F1 and F2 with 2–3 and 4–5 longitudinal placodes, respectively*Lipolexis pakistanicus* sp. n.


## 4. Discussion

Until recently, taxonomists relied on morphological and ecological studies as a basis for describing new species or identifying newly collected specimens. However, easy access to DNA sequence information has now enabled integrative taxonomy which combines morphological/ecological and molecular data to clarify species boundaries. The barcode region of COI is by far the most widely utilized molecular marker used for molecular studies on the subfamily Aphidiinae [30,44,45]. Aside from aiding the identification of the newly collected specimens, this gene region aids the delineation of species boundaries, an approach of particular value for aphidiine genera that are suspected to include cryptic species complexes [46,47,48,49]. Our study reinforced the value of COI as it separated *Lipolexis* species with high resolution and bootstrap support. The molecular results were concordant with those from morphological study leading to the description of six new species of *Lipolexis*. Each of these species possessed diagnostic morphological characters which separates it from its congeners which makes the fact that they went unnoticed surprising.

Based on morphological and molecular differences, *Lipolexis* can be separated into two main groups, i.e., *gracilis* and *oregmae*. The average genetic distance between species in these two groups is high, ranging from 21.7% to 23.2%. Furthermore, *gracilis* and *oregmae* groups are morphologically distinguished by the differing shape of the petiole: members of the *L. gracilis* group have a petiole with prominent central carinae, while those in the *L. oregmae* group have a petiole that is smooth dorsally, but with noticeably crenulated lateral longitudinal carinae. Additionally, members of the *gracilis* group have a slightly shorter metacarpal vein (R1) than members of the *oregmae* group (proportion between length of pterostigma and metacarpal vein is 0.90–1.1 in *gracilis* group versus 0.75–0.90 in *oregmae* group).

Five new species belonging to the *L. gracilis* group were discovered in this study: *Lipolexis labialis* sp. n., *L. pelopsi* sp. n., *L. takadai* sp. n., *L. pakistanicus* sp. n. and *L. peregrinus* sp. n. Furthermore, within the *gracilis* group, two additional clusters are formed, one constituted of *L. gracilis* s. str. and *L. labialis* sp. n., and the second of the remaining species (*L. pelopsi* sp. n., *L. takadai* sp. n., *L. pakistanicus* sp. n. and *L. peregrinus* sp. n.) with the average between group distance of 11.4%. Although *L. labialis* sp. n. is most closely related to *L. gracilis* (average genetic distance of 7.3%), the two species are readily distinguished by their differing number of labial palpomeres. *L. gracilis* possesses one labial palpomere while *L. labialis* sp. n. has two. These two species have overlapping distributions in Europe, but may use different hosts. All reared specimens of *L. labialis* employed Macrosiphini aphid hosts (*Myzus* Passerini, *Roepkea* Hille Ris Lambers) excepting one reared from *Anoecia corni* (F.). Although *L. gracilis* has also been reported from *Myzus* [43,50], in the light of the new species discoveries, it is not clear whether the specimens reared from *Myzus* are indeed *L. gracilis* or might be some other *Lipolexis* species, such as *L. labialis* sp. n. or *L. peregrinus* sp. n.

In this phylogenetic analysis, within the second cluster of the *gracilis* group, *L. takadai* sp. n. wasrepresented by just a single specimen from China. However, morphological examination of other *Lipolexis* revealed two specimens from Japan which fully correspond to it. Although molecular data indicate that *L. takadai* sp. n. is most closely related to *L. peregrinus* sp. n., *L. pakistanicus* sp. n. and *L. pelopsi* sp. n. (3.9%, 4.5%, 6.8% average COI divergence respectively), it possesses several morphological differences from other species of *Lipolexis*. Although it is only known from China and Japan, its actual distribution may be much broader. *Lipolexis takadai* sp. n. was reared from *Aphis gossypii* Glover, a common host for other *Lipolexis*.

A second species from this cluster, *Lipolexis pelopsi* sp. n., is most closely related to *L. peregrinus* sp. n., *L. pakistanicus* sp. n. and *L. takadai* sp. n., with 6.9%, 8.6%, 6.8% average COI distance, respectively. It has a Mediterranean distribution as it was reared from aphids collected in Greece, Bosnia and Herzegovina, Croatia, and Montenegro. It is a parasitoid of aphids in the genus *Aphis* attacking various plant species. *Lipolexis pelopsi* sp. n. coccurs in this region with *L. gracilis* and *L. labialis* sp. n., and also shares an aphid host (*Aphis* sp.) with *L. gracilis*. It is characterised by a pubescent body (specimens from Greece possess heavily setose mesoscutum), a feature that distinguishes it from all other species.

*Lipolexis peregrinus* sp. n., another species in the same cluster, is most closely related to *L. pakistanicus* sp. n., *L. takadai* sp. n. and *L. pelopsi* sp. n. (2.2%, 3.9%, 6.9% average COI distance, respectively). Its specimens were collected in both Asia (Japan, China) and Europe (Spain, Slovenia). Further investigations are needed to ascertain if *L. peregrinus* sp. n., is present across the Palaearctic or if it was introduced from Asia to Europe or vice versa. Given its phylogenetic position within the *gracilis* group and its number of palpomeres, it probably originates from the Oriental region. *Lipolexis peregrinus* sp. n. is phylogenetically closest to Asian species and its maxillary palps have 3 palpomeres, a trait that characterizes Asian species, while the European species (i.e., *L. gracilis*, *L. pelopsi* sp. n. and *L. labialis* sp. n.) possess maxillary palps with 4 palpomeres. In Asia, *L. peregrinus* sp. n. was found to parasitize aphid genera *Aphis* L., *Melanaphis* and *Toxoptera* Koch [51,52] (in both studies identified as *L. gracilis*), while in Europe it was reared from *M. persicae* and *Aphis* sp.

*Lipolexis pakistanicus* sp. n. is phylogenetically most closely related to *L. peregrinus* sp. n., *L. pelopsi* sp. n. and *L. takadai* sp. n. (2.2%, 8.6%, 4.5% average COI distances, respectively). With the exception of two specimens from Bangladesh and two from Moldova, all material was collected in Pakistan. As most specimens were collected in Malaise traps, the aphid host spectrum is unknown, but one specimen from Pakistan was reared from *A. gossypii*. This species is likely distributed across the Oriental region. In Bangladesh, it co-occurs with *L. oregmae* and *L. bengalensis* sp. n., but more information is needed on host use to ascertain the extent of overlap.

Although *L. bengalensis* sp. n. is a member of the *oregmae* group, based on both sequence results and morphology, it is genetically distant from *L. oregmae* (19.9% average), implying the species have long been separate. At the moment, *L. bengalensis* sp. n. is only known from Bangladesh where it was collected in a vegetable crop field where *L. oregmae* was also collected, indicating their co-occurence. Its aphid host is unknown because it was collected by Malaise trap.

Starý [53] reported that *L. gracilis* is common in steppe habitat, orchards, and wood edges. Our material of *L. gracilis* originates from high montane habitat, mixed forests, steppes, orchards, and urban area. Although *L. gracilis* mostly parasitizes hosts from the tribes Aphidini (genus *Aphis* Linnaeus) and Macrosiphini (genera *Brachycaudus* and somewhat less *Myzus* Passerini), it has been reared from other unrelated aphid genera, such as *Anoecia* Koch, *Myzocallis* and *Therioaphis* [43]. By comparison, the Oriental *L. oregmae* is mostly associated with aphids from the tribes Aphidini (genera *Aphis*, *Rhopalosiphum* Koch and *Toxoptera* Koch) and Macrosiphini (genera *Cavariella*, *Liosomaphis*, *Myzus*, *Pentalonia*, *Semiaphis* and *Sitobion*), but it also attacks representatives of some tribes ignored by *L. gracilis* such as Cerataphidini (*Ceratovacuna lanigera* Zehntner), Greenideini (*Greenidea psidii* van der Goot (=*formosana* (Maki)) and Tuberolachnini (*Tuberolachnus salignus* (J. F. Gmelin)) [43]. However, in light of our discovery of six new species, host use might not be that broad as previously thought so trophic associations for these two species require careful validation.

With the exception of *L. labialis* sp. n., all other species of *Lipolexis* parasitize *Aphis* despite their sympatry in the same habitat. For instance, in Europe, both *L. gracilis* and *L. labialis* sp. n. parasitize *Aphis* spp. Furthermore, three species were found in Montenegro, i.e., *L. labialis* sp. n., *L. pelopsi* sp. n. and *L. gracilis*, while, *L. labialis* sp. n., *L. gracilis* and *L. peregrinus* sp. n. were recorded from Slovenia. As a result, it seems clear that speciation in the genus *Lipolexis* has not been driven by host specialization.

Several studies have reported unusually high COI distances between species within the subfamily Aphidiinae [47,49]. For example Kocić et al. [47] found COI average distances of up to 20.7% among species in the phylogenetically old aphidiine genus *Ephedrus*. Similarly, some species of *Lipolexis* showed even higher average distances (i.e., 23.2% between *L. bengalensis* sp. n. and *L. peregrinus* sp. n.). While adults of *Ephedrus* possess several plesiomorphic taxonomic characters, this is not the case for *Lipolexis*. In fact, *Lipolexis* is characterized by several apomorphic taxonomic characters, such as reduced wing venation, elongated petiole, and shape of the ovipositor sheath. Its apomorphies and phylogenetic studies at the subfamily level (where it is clustered with evolutionary younger genera *Binodoxys* and *Trioxys*) suggest this genus evolved quite recently. However, the average genetic distances of COI, which are even higher than those in *Ephedrus*, might indicate that *Lipolexis* diversification was not a recent event. Furthermore, Finlayson [25] stated that the larval morphology of *Lipolexis* exhibits plesiomorphic morphological characters. One possibility might be that *Lipolexis* separated early in the evolution of Aphidiinae and acquired adult apomorphic traits independently. Further research is needed in order to try to resolve the complex and poorly investigated position of *Lipolexis* and its relationships with the other aphidiine members.

## 5. Conclusions

Our study revealed six new species of *Lipolexis* (*L. peregrinus* sp. n., *L. pelopsi* sp. n., *L. labialis* sp. n., *L. takadai* sp. n., *L. pakistanicus* sp. n. and *L. bengalensis* sp. n.). Each of these species possesses clear morphological characters that distinguishes it from its congeners. Moreover, two groups can be differentiated within the genus—*oregmae* and *gracilis*. The genetic distances between them are even higher than intrageneric distances within the potentially genus *Ephedrus*. The *oregmae* group consists out of *L. oregmae* and *L. bengalensis* sp. n. of Oriental origin. The *gracilis* group of species can be separated into two clades, one consisting of *L. gracilis* and *L. labialis* sp. n., and the other of *L. peregrinus* sp. n., *L. takadai* sp. n., *L. pelopsi* sp. n. and *L. pakistanicus* sp. n. Furthermore, the consistency of number of palpomeres is in concordance with the geographical distribution: the European species (*L. gracilis*, *L. labialis* sp. n. and *L. pelopsi* sp. n.) possess four maxillar palpomeres, while the Oriental ones (*L. oregmae*, *L. takadai*, *L. bengalensis*, *L. peregrinus* and *L. pakistanicus*) have three. We provide a key to the world *Lipolexis* species.

## Figures and Tables

**Figure 1 insects-11-00667-f001:**
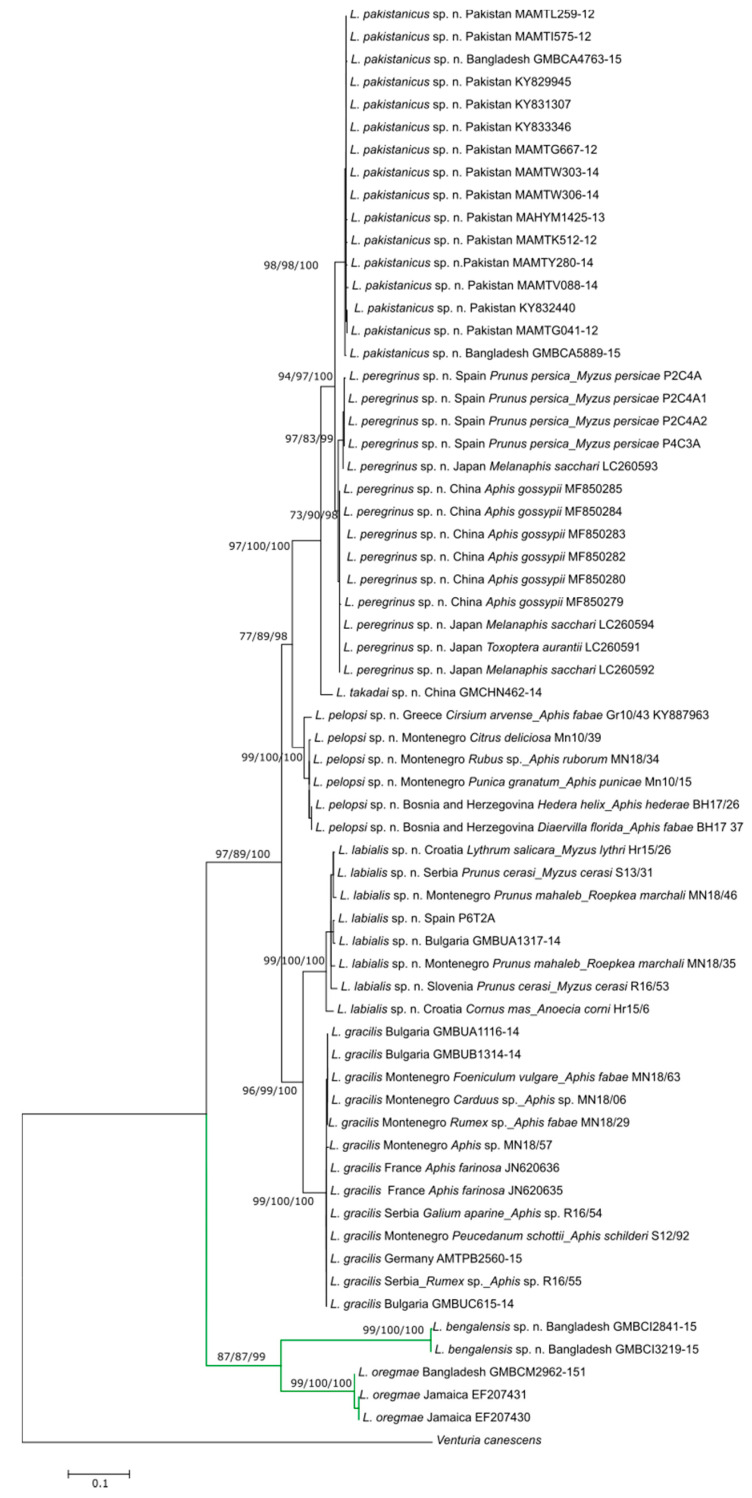
Phylogenetic tree based on 63 cytochrome oxidase *c* subunit I (COI) sequences of *Lipolexis*. obtained using Neighbor Joining analysis. Bootstrap values/Bayesian posterior probabilities above 50% are shown on the branches in ML/NJ/Bayesian analysis order. Specimen data are presented in following array: parasitoid/country of origin/aphid and plant host (if available)/BOLD, GenBank or private code. Green branches represent *L. oregmae* lineage, black *L. gracilis*.

**Figure 2 insects-11-00667-f002:**
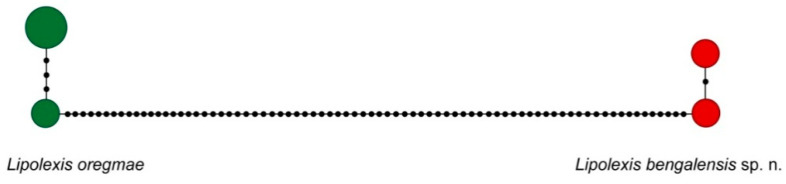
Haplotype network of COI sequences obtained from specimens in the *L. oregmae group*. The circle size indicates the number of specimens with a particular haplotype; each black dot represents a nucleotide substitution.

**Figure 3 insects-11-00667-f003:**
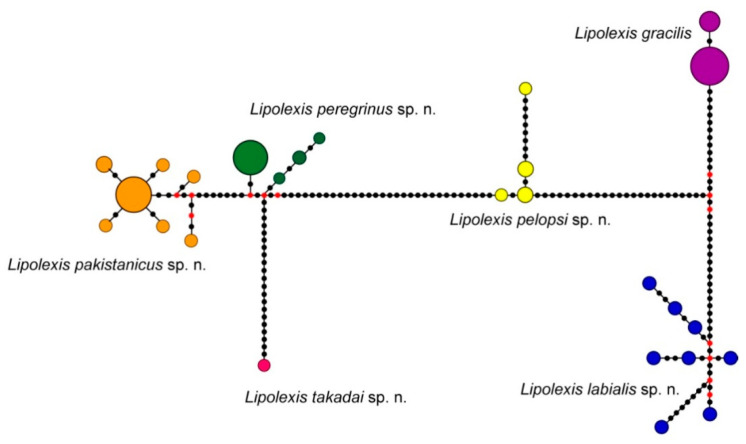
Haplotype network for COI sequences from 58 specimens belonging to the *L. gracilis* group. The circle size indicates the number of specimens with a haplotype; each black dot represents a nucleotide substitution, red median vectors.

**Figure 4 insects-11-00667-f004:**
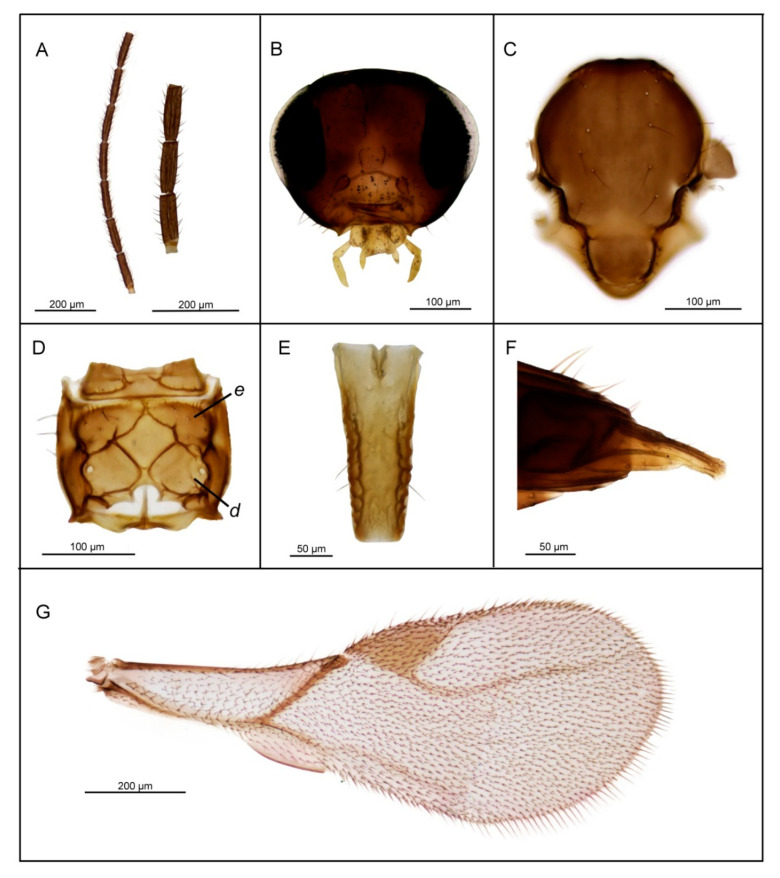
*Lipolexis bengalensis* sp. n., female: (**A**) antenna and F1–F3; (**B**) head; (**C**) mesoscutum; (**D**) propodeum; (**E**) petiole, dorsal view; (**F**) ovipositor sheath; (**G**) forewing; ***e*** external areola, ***d*** dentiparal areola.

**Figure 5 insects-11-00667-f005:**
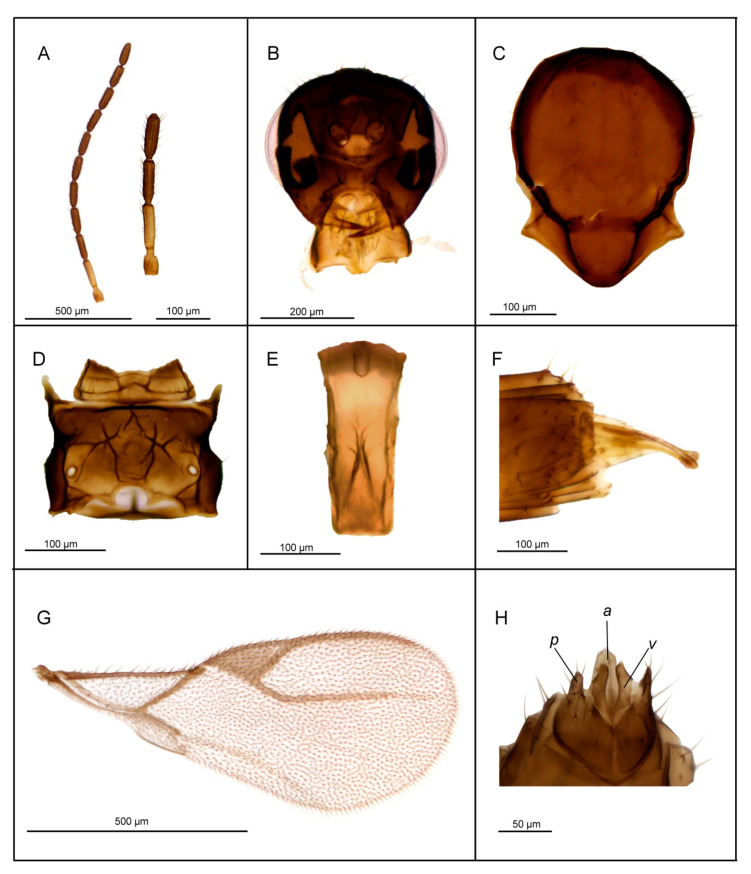
*Lipolexis labialis* sp. n.: female (**A**) antenna and F1–F3; (**B**) head; (**C**) mesoscutum; (**D**) propodeum; (**E**) petiole, dorsal view; (**F**) ovipositor sheath; (**G**) forewing; male (**H**) aedeagus; ***p*** paramere, ***a*** aedeagus, ***v*** volsella.

**Figure 6 insects-11-00667-f006:**
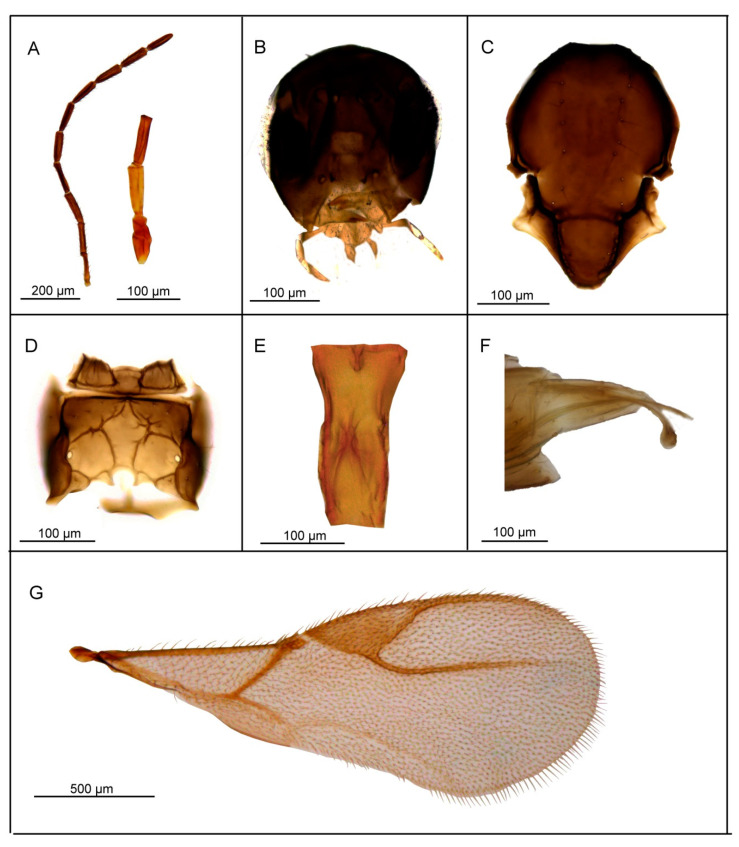
*Lipolexis takadai* sp. n. female: (**A**) antenna and F1 and F2; (**B**) head; (**C**) mesoscutum; (**D**) propodeum; (**E**) petiole, dorsal view; (**F**) ovipositor sheath; (**G**) forewing.

**Figure 7 insects-11-00667-f007:**
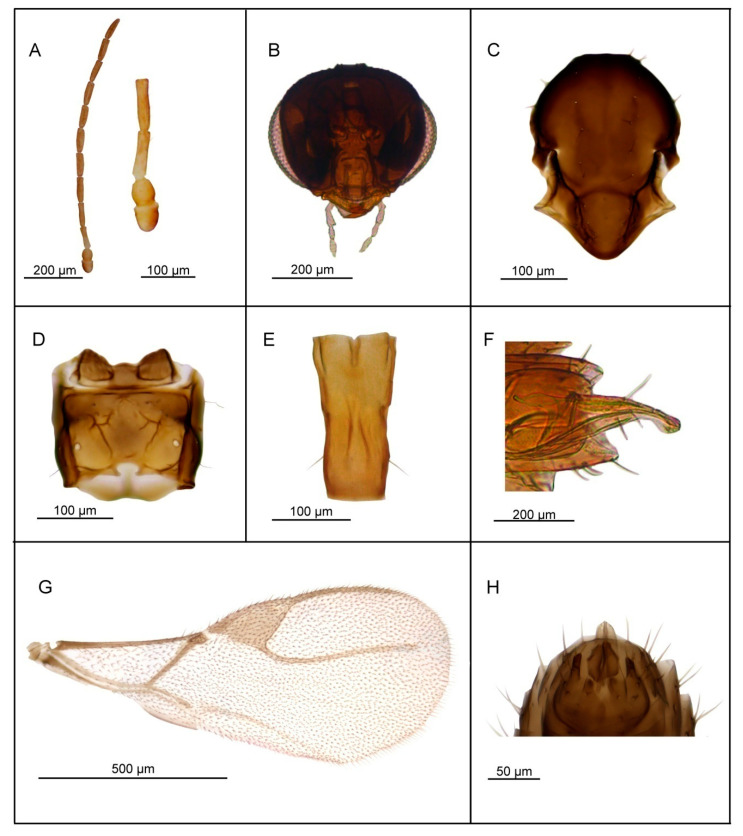
*Lipolexis pelopsi* sp. n.: female (**A**) antenna and F1 and F2; (**B**) head; (**C**) mesoscutum; (**D**) propodeum; (**E**) petiole, dorsal view; (**F**) ovipositor sheath; (**G**) forewing; male (**H**) aedeagus.

**Figure 8 insects-11-00667-f008:**
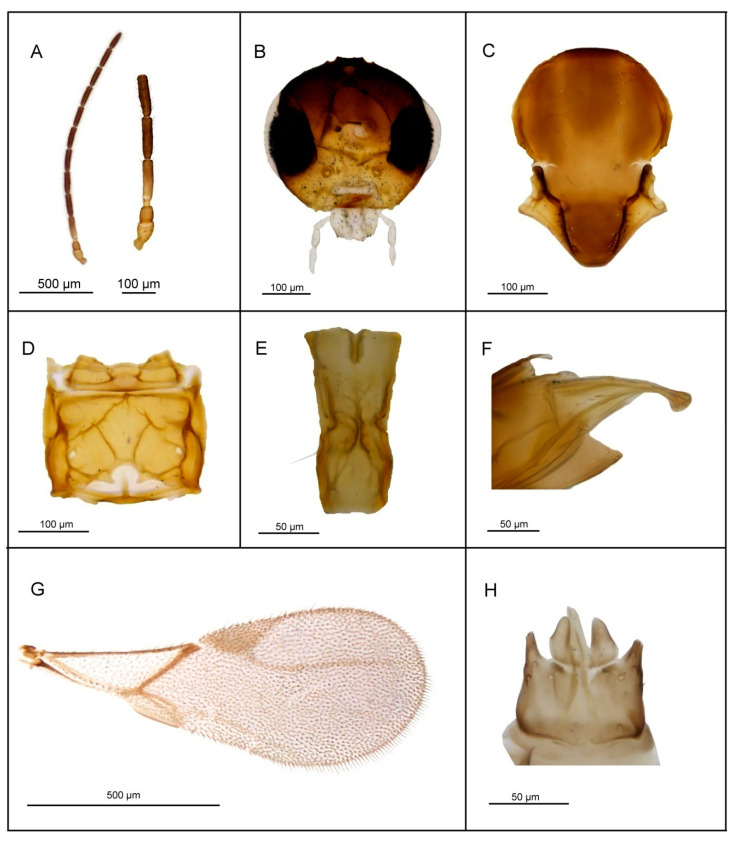
*Lipolexis pakistanicus* sp. n.: female (**A**)antenna and F1–F3; (**B**) head; (**C**) mesoscutum; (**D**) propodeum; (**E**) petiole, dorsal view; (**F**) ovipositor sheath; (**G**)forewing; male (**H**) aedeagus.

**Figure 9 insects-11-00667-f009:**
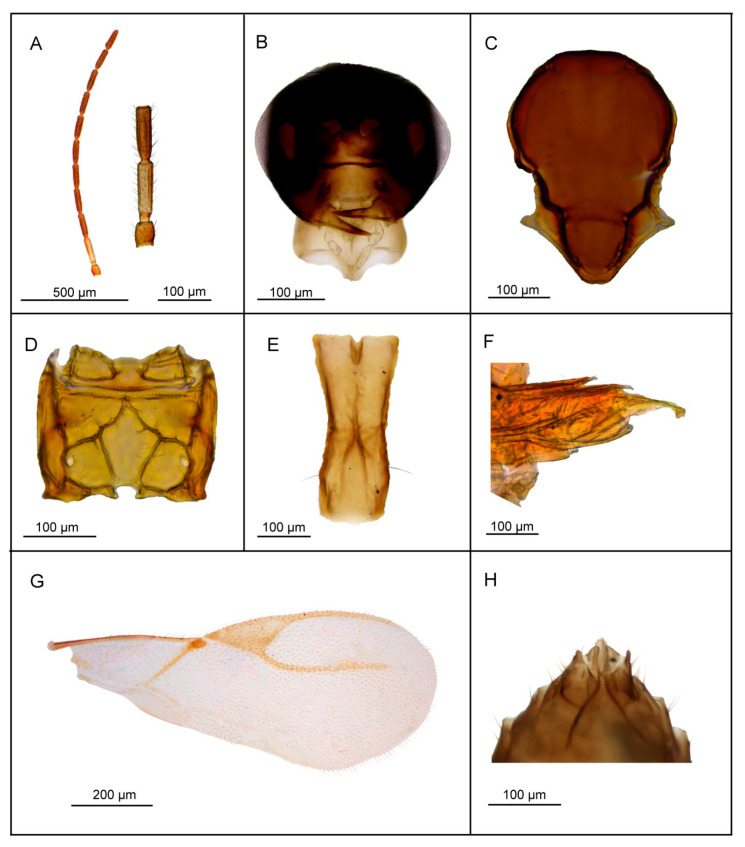
*Lipolexis peregrinus* sp. n.: female (**A**) antenna and F1 and F2; (**B**)head; (**C**) mesoscutum; (**D**) propodeum; (**E**) petiole, dorsal view; (**F**) ovipositor sheath; (**G**) forewing; male (**H**) aedeagus.

**Figure 10 insects-11-00667-f010:**
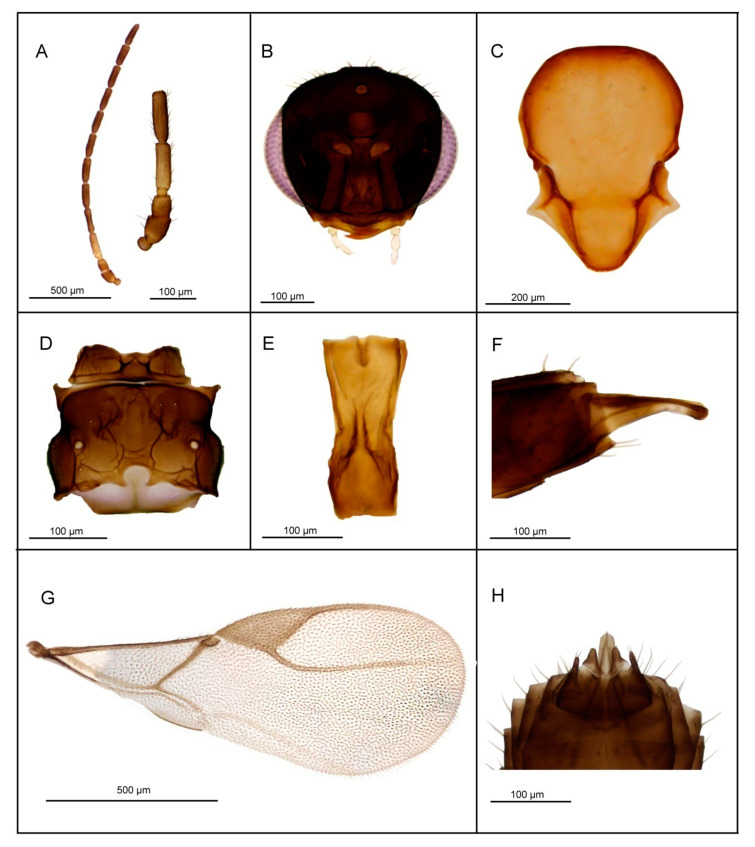
*Lipolexis gracilis:* female (**A**) antenna and F1 and F2; (**B**) head; (**C**) mesoscutum; (**D**) propodeum; (**E**) petiole, dorsal view; (**F**) ovipositor sheath; (**G**) forewing; male (**H**) aedeagus.

**Figure 11 insects-11-00667-f011:**
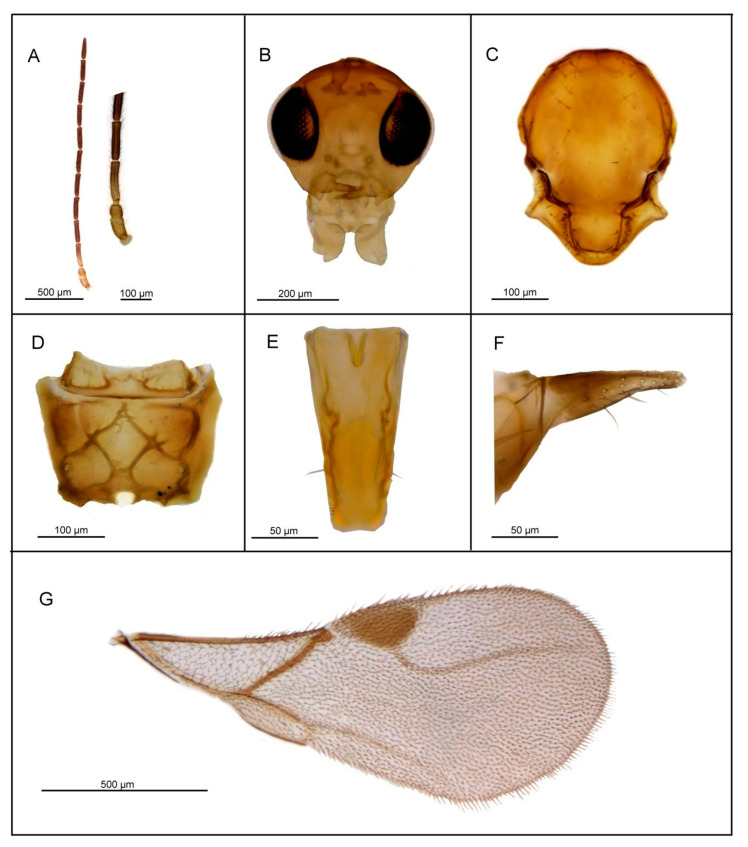
*Lipolexis oregmae*: (**A**) antenna and F1–F3; (**B**) head; (**C**) mesoscutum; (**D**) propodeum; (**E**) petiole, dorsal view; (**F**) ovipositor sheath; (**G**) forewing.

**Table 1 insects-11-00667-t001:** Average genetic distances for the COI gene between eight *Lipolexis* species.

Species	*L. peregrinus* sp. n.	*L. gracilis*	*L. pelopsi* sp. n.	*L. labialis* sp. n.	*L. pakistanicus* sp. n.	*L. takadai* sp. n.	*L. bengalensis* sp. n.
*L. peregrinus* sp. n.							
*L. gracilis*	0.111						
*L. pelopsi* sp. n.	0.069	0.086					
*L. labialis* sp. n.	0.123	0.073	0.097				
*L. pakistanicus* sp. n.	0.022	0.110	0.086	0.132			
*L. takadai* sp. n.	0.039	0.097	0.068	0.106	0.045		
*L. bengalensis* sp. n.	0.232	0.217	0.227	0.228	0.231	0.229	
*L. oregmae*	0.206	0.215	0.218	0.232	0.213	0.200	0.199

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
