# Peer review of "Resolving the Taxonomic Status of Potential Biocontrol Agents Belonging to the Neglected Genus Lipolexis Förster (Hymenoptera, Braconidae, Aphidiinae) with Descriptions of Six New Species"

_insects, 2020, doi:10.3390/insects11100667_

Round 1

Reviewer 1 Report

This manuscript is an interesting insight into the systematics and classification of the braconid genus Lipolexis. Although some parts of the manuscript are well written and generally this study is worth to be published, there are some parts which need to be improved.

Because the aticle brings new taxonomic acts (new species names), it MUST be registered in ZooBank in order to be valid and meet the criteria of the ICZN (because Insects is an online journal). This is not a recommendation, it is a must! (otherwise the names would not be valid according to the principles of ICZN)

Please, provide LSID numbers for the article as well as for all new species.

Title

As it is, the title is appropriate for some taxonomic journal like Zootaxa or ZooKeys. Because Insects is more prestigious journal focused on more general problems I would highly recommend using the word "pest" and/or "DNA barcoding" and/or "cryptic species" or something more catchy to attract wider audience and show that this article touches also some general topic and not only pure taxonomy. This would be good for both authors as well as for the journal itself because it can generate more citations.

Introduction

- I would be happy if there is some explanation about the recent classification of the genus - in which tribe, subtribe, how many tribes are in the subfamily, etc.. non-experts can easily be lost later when reading other parts of the manuscript. Please add this.

- I recommend adding information about the ecology/life-history of this genus into Introduction, for example that they are parasitioids and of which insects, etc.. Reader can find this in the middle of the manuscript only, which is not ideal. I would recommend adding this information to Introduction, maybe on the beginning as an introductory sentence (this would also enable readers to understand why this taxon is worth of further study)

- line 50: "a third" should be "the third"

- lines 57-64: this part should be removed from Introduction and placed to Methods or Results because these are, yet unpublished, hypotheses of authors. What is more, this is important for other discussion, and also for the circumscription of the genus itself, and if you consider these genera synonyms, these synonymies should definitely be placed under the respective species (oregmae) in the overview/redescription of the species (synonyms missing there)...even if you only hypothesize this, this remark should be placed under L. oregmae and not in Introduction (in Introduction only bare facts that these species were described by the Chinese/Indian authors and their status is doubtful ("see Comment under L. oregmae"; or similarly) should be written)

- line 60: "doesn't" should be "does not"

- line 66: comma after "species"

- line 69: comma after "oregmae" should be removed

- lines 77-82: subtribe Trioxina..but of which tribe and subfamily? If this was explained earlier in Introduction (see one of my previous comments on the classification of the genus and subfamily), readers could better understand this part

- line 103: space between USA and (Florida)

- line 106: missing word (e.g., various?) after "from"

Material and Methods

°C - maybe it is because of the font used, but this character is strange

- line 119: extra space between "light/" and "8"

- line 122: extra comma after "stereomicroscopes"

Here comes one of my major criticism: this part should be improved in order the readers know more details, and consequently, information on the dataset and analyses are straightforward..

For example, there should be clearly written How many samples does the final dataset contain, How many of these samples were newly sequenced and how many of these were added from GenBank/BOLD, and these from GenBank/BOLD were only Lipolexis sequences or also some other (outgroup)? (64 taxa, 63 Lipolexis, 23 newly sequenced? etc.) This information can be found at three various places in Methods and also Results section, but this should be clearly stated at one place in Methods. Furthermore, the analyses settings should be better explained. How many million generations were used for the Bayesian analysis? Were the data partitioned for different codon positions? If not, why? If so, did you use any program to test which partition strategy is the best for your data? What about the saturation of the third codon positions? I have an experience that they are very often saturated and should be discarded from the dataset, and that the analyses of datasets with and without the third codon positions may be quite different... This should be explored, for example in the DAMBE software.

Results

line 153: "COI sequences were recovered from 63 specimens.." this sounds like you newly recovered these sequences; however, many of them were downloaded from GenBank/BOLD

line 161: - between numbers

One tense (past) tense should be used. For example, on line 188 there is "is"..

line 192: "readers are referred for a phylogenetic tree to Fig. 3; however, figure 3 is not a phylogenetic tree!

line 195: "The balance of the Results section..." what does this mean?

Species descriptions are well-prepared and are supplemented by useful set of photographs of main diagnostic characters. However, there are some places where italics is needed for genus/species names (e.g., lines 201, 244, 299, 357, 471, 532), and some misspellings should be checked (testaceous vs testaceus, 13 - segmented vs 13-segmented, space between Lipolexis and chinensis, extra space on line 431, etc.).

It is usually better to write "antenna with 13 antennomeres", because some authors consider antennae are only of three segments, i.e., scape, pedicel and flagellum, formed by flagellomeres..

Generally, all species should have the "same form", for example, please add also the "Host" subsection for all species including not only the new ones (missing in two already described species), add the "Diagnosis" sections also for two already described species, gracilis and oregmae. This is really important so that all species treated here have clearly written Diagnosis.

Is it really OK to use the name "pelopsi" instead of "peloponnensis", which should be the correct formation of name based on the Peloponnese?? As it is, it sounds me odd although I am not expert in forming new names according to the principles of the ICZN. Please, double-check this and consider using "peloponnensis".

line 433: what are these numbers (2.9, etc.)? some lengths in mm, or ratios? of what? please, explain in the manuscript.

Please add also author and publication, page etc. for L. oregmae (added for gracilis but not here!), and also include synonyms, if any... what about the Chinese/Indian species? this is a great opportunity to place there a note about these species, or possibly make formal synonymies (in this case they should appear in Abstract also). What about L. scutellaris etc.? Did you study the type series? Can you confirm the synonymy made by Stary?

Please, write here ALL species names, either some clear notes, or as synonyms, but it is not possible to make this review incomplete (!)

Discussion

lines 682-683: Please check this sentence and rewrite in order it makes sense.

line 684: replace "study" by "analysis"

line 686: correspond "with" it (?)

line 693: an extra space

line 696: The new sentence should not start with the abbreviated Latin name, use it in full in this case

line 694: "possesses...distribution" sounds odd to me (maybe "has" would be better?)

line 731: "attacks some tribes" - please rewrite, it does not attack tribes but "the representatives/members of tribes..."

line 744 and in other places: Please, do NOT use the "possibly basal" or "basal" for the genus Ephedrus... I know what you want to say but this is not a good expression. Furthermore, was such position suggested by any phylogenetic analysis or this is just a hypothesis made by morphologists? The same for using "primitive" characters...what does it mean? I would avoid using these terms.

Authos Contributions

Some authors are not explicitely mentioned here, although I assume they all "reviewed and recised the paper" which was their role and this is why they are the co-authors and not in Acknowledgements.

Author Response

Reviewer1

We would like to thank Reviewer 1 for the constructive comments and suggestions that will improve our manuscript.

Comment 1.

Because the aticle brings new taxonomic acts (new species names), it MUST be registered in ZooBank in order to be valid and meet the criteria of the ICZN (because Insects is an online journal). This is not a recommendation, it is a must! (otherwise the names would not be valid according to the principles of ICZN)

Please, provide LSID numbers for the article as well as for all new species.

At the moment, the ZooBank webpage is not working, so it is not possible to provide LSID numbers. We will do it in due time.

Comment 2.

As it is, the title is appropriate for some taxonomic journal like Zootaxa or ZooKeys. Because Insects is more prestigious journal focused on more general problems I would highly recommend using the word "pest" and/or "DNA barcoding" and/or "cryptic species" or something more catchy to attract wider audience and show that this article touches also some general topic and not only pure taxonomy. This would be good for both authors as well as for the journal itself because it can generate more citations.

The title of the manuscript is changed to: “Resolving the taxonomic status of potential biocontrol agents belonging to the neglected genus Lipolexis Förster (Hymenoptera, Braconidae, Aphidiinae) with descriptions of six new species”

Comment 3.

Introduction

- I would be happy if there is some explanation about the recent classification of the genus - in which tribe, subtribe, how many tribes are in the subfamily, etc.. non-experts can easily be lost later when reading other parts of the manuscript. Please add this.

Unfortunately, there is no wide accepted classification of the Aphidiinae species on the tribe and subtribe level.

Line 83: sentence added: This subtribe is classified by different authors within tribe Trioxini [23] or tribe Aphidiini [22].

Comment 4.

- I recommend adding information about the ecology/life-history of this genus into Introduction, for example that they are parasitioids and of which insects, etc.. Reader can find this in the middle of the manuscript only, which is not ideal. I would recommend adding this information to Introduction, maybe on the beginning as an introductory sentence (this would also enable readers to understand why this taxon is worth of further study)

line 42– Sentence added: Like all members of this subfamily, species of Lipolexis are solitary koinobiont endoparasitoids of aphids (Aphidiidae).

Comment 5.

- line 50: "a third" should be "the third"

Changed.

Comment 6.

- lines 57-64: this part should be removed from Introduction and placed to Methods or Results because these are, yet unpublished, hypotheses of authors. What is more, this is important for other discussion, and also for the circumscription of the genus itself, and if you consider these genera synonyms, these synonymies should definitely be placed under the respective species (oregmae) in the overview/redescription of the species (synonyms missing there)...even if you only hypothesize this, this remark should be placed under L. oregmae and not in Introduction (in Introduction only bare facts that these species were described by the Chinese/Indian authors and their status is doubtful ("see Comment under L. oregmae"; or similarly) should be written)

The remarks concerning the validity of these three species moved to the results section, as suggested by the reviewer

lines 628-636: Remarks about L. wuyiensis, L. pseudosctullearis and L. myzakkaiae. On the basis of its description, L. wuyiensis is very similar to L. oregmae and it parasitizes Ceratovacuna lanigera Zehntner (=Oregma lanigera), the same aphid host as L. oregmae in the Philippines. Furthermore, apart from the drawing of the petiole of L. myzakkaiae, which does not follow the original description (i.e., two lateral longitudinal wavy branched carinae), both L. myzakkaiae and L. pseudoscutellaris are morphologically similar to L. oregmae. Moreover, it seems that neither species has been mentioned in the scientific literature since their description. In our opinion, all three species are likely to be conspecific with L. oregmae. However, due the absence of type material, the authors refrained from official synonymization

Comment 7

  • Line 60: “doesn’t” should be “does not”

Changed.

Comment 8

  • Line 66: comma after species

Added.

Comment 9.

  • Line 69: comma after “oregmae” should be removed

Removed.

Comment 10.

- lines 77-82: subtribe Trioxina..but of which tribe and subfamily? If this was explained earlier in Introduction (see one of my previous comments on the classification of the genus and subfamily), readers could better understand this part

Please see answer in the Comment 3.

Comment 11.

- line 103: space between USA and (Florida)

There is a space between USA and Florida.

Comment 12.

- line 106: missing word (e.g., various?) after "from"

Word “various” added.

Comment 13

°C - maybe it is because of the font used, but this character is strange

  • line 119: “°” changed to different font type ()

Comment 14.

- line 119: extra space between "light/" and "8"

- line 120: extra space deleted.

Comment 15.

- line 122: extra comma after "stereomicroscopes"

- line 123: extra comma deleted.

Comment 16.

Here comes one of my major criticism: this part should be improved in order the readers know more details, and consequently, information on the dataset and analyses are straightforward..

For example, there should be clearly written How many samples does the final dataset contain, How many of these samples were newly sequenced and how many of these were added from GenBank/BOLD, and these from GenBank/BOLD were only Lipolexis sequences or also some other (outgroup)? (64 taxa, 63 Lipolexis, 23 newly sequenced? etc.) This information can be found at three various places in Methods and also Results section, but this should be clearly stated at one place in Methods.

lines 158-160. Sentence edited to summarize the sequences used in the analyses: The final dataset contained 64 COI sequences (one outgroup sequence, additional 40 sequences acquired from BOLD systems database (25) and GenBank (15), and 23 newly recovered sequences.

Furthermore, the analyses settings should be better explained. How many million generations were used for the Bayesian analysis? Were the data partitioned for different codon positions? If not, why? If so, did you use any program to test which partition strategy is the best for your data? What about the saturation of the third codon positions? I have an experience that they are very often saturated and should be discarded from the dataset, and that the analyses of datasets with and without the third codon positions may be quite different... This should be explored, for example in the DAMBE software.

Added the number of generations, burn in, sample frequency, checked for ess values. The data was in one case partitioned in two ways (1+2+3, and (1+2)3)) and in the second not, and results were the same. We checked for the saturation of the third codon position in DAMBE software and they were not saturated.

Lines 151-155. Added: The analysis ran for ten million generations, the sampling was conducted every 1000 generations, while the first million of trees was discarded as a burn-in. The effective sample size (ESS) of the parameters of the Markov chain Monte Carlo was estimated by Tracer v1.7.1 (36).The saturation level for the third codon position was inspected in DAMBE software (37).

Comment 17.

line 153: "COI sequences were recovered from 63 specimens.." this sounds like you newly recovered these sequences; however, many of them were downloaded from GenBank/BOLD

line 164: Changed to "COI sequences were acquired from 63 specimens"

Comment 18.

line 161: - between numbers

Changed to “–“.

Comment 19

One tense (past) tense should be used. For example, on line 188 there is "is"

Changed to past tense in lines 194 and 195.

Comment 20

line 192: "readers are referred for a phylogenetic tree to Fig. 3; however, figure 3 is not a phylogenetic tree!

line 198: Readers were supposed to be referred to the haplotype network (Fig 3). The referral was moved to the first part of the sentence in order to be more specific

Comment 21

line 195: "The balance of the Results section..." what does this mean?

Means: the summary

Comment 22.

Species descriptions are well-prepared and are supplemented by useful set of photographs of main diagnostic characters. However, there are some places where italics is needed for genus/species names (e.g., lines 201, 244, 299, 357, 471, 532), and some misspellings should be checked (testaceous vs testaceus, 13 - segmented vs 13-segmented, space between Lipolexis and chinensis, extra space on line 431, etc.).

Italics changed, misspellings checked, spaces deleted throughout the manuscript.

Comment 23.

It is usually better to write "antenna with 13 antennomeres", because some authors consider antennae are only of three segments, i.e., scape, pedicel and flagellum, formed by flagellomeres..

The authors are aware that some authors consider antennae to be built out of three segments, but within Aphidiinae subfamily, this terminology is widely accepted.

Comment 24.

Generally, all species should have the "same form", for example, please add also the "Host" subsection for all species including not only the new ones (missing in two already described species), add the "Diagnosis" sections also for two already described species, gracilis and oregmae. This is really important so that all species treated here have clearly written Diagnosis.

Added sections Hosts and Diagnosis for both L. gracilis and L. oregmae

Lines 562-566 Diagnosis. Lipolexis gracilis has a petiole with central bifurcating carinae, a feature that positions it within the gracilis species group. It possesses maxillary palps with four palpomeres, labial palps with one palpomere (the same combination as L. pelopsi). However, it differs from the latter by less pubescent body and elongated petiole (proportions between length and maximum width of petiole at spiracles, 3.0–3.6 in L. gracilis vs. 2.4–2.6 in L. pelopsi sp. n.).

Lines 608-611 Hosts. Acyrthosiphon Mordvilko, Aphis L., Brachycaudus van der Goot, Capitophorus van der Goot, Liosomaphis Walker, Lipaphis Mordvilko, Melanaphis van der Goot, Metopeurum Mordvilko, Myzocallis Passerini, Myzus, Rhopalosiphum, Semiaphis van der Goot, Therioaphis Walker and Toxoptera.

Lines 626-629 Diagnosis. Lipolexis oregmae possesses a petiole that bears lateral longitudinal carinae, which is a morphological character of the oregmae group of species. It differs from the other member of the group, L. bengalensis sp. n. by the number of maxillary palpomeres (three maxillar palpomeres in L. oregmae vs, two in L. bengalensis sp. n.)

 Lines 654-656 Hosts. Aphis, Cavariella del Guercio, Ceratovacuna Zehtner, Greenidea Schouteden, Liosomaphis, Myzus, Pentalonia Coquerel, Rhopalosiphum, Semiaphis, Sitobion Mordvilko, Toxoptera, Tuberolachnus Mordvilko.

Comment

Is it really OK to use the name "pelopsi" instead of "peloponnensis", which should be the correct formation of name based on the Peloponnese?? As it is, it sounds me odd although I am not expert in forming new names according to the principles of the ICZN. Please, double-check this and consider using "peloponnensis".

line 421-422 Etimology explained in more detail: Lipolexis pelopsi sp. n. takes its name after Pelops, mythological king of Peloponnese region, which is the area of the first collected specimens.

Comment 25.

line 433: what are these numbers (2.9, etc.)? some lengths in mm, or ratios? of what? please, explain in the manuscript

line 439: added “length to width ratio”

Comment 26

Please add also author and publication, page etc. for L. oregmae (added for gracilis but not here!), and also include synonyms, if any... what about the Chinese/Indian species? this is a great opportunity to place there a note about these species, or possibly make formal synonymies (in this case they should appear in Abstract also). Please, write here ALL species names, either some clear notes, or as synonyms, but it is not possible to make this review incomplete (!)

lines 625-626 Recognized synonyms added: Diaeretus oregmae (Gahan, 1932), Ann Entomol Soc Am, 25, 736-757; Lipolexis scutellaris Mackauer, 1962, Entomophaga, 7, 1, 37-45

line 627 Moved from the Introduction section: Remarks about L. wuyiensis, L. pseudosctullearis and L. myzakkaiae. On the basis of its description, L. wuyiensis is very similar to L. oregmae and it parasitizes Ceratovacuna lanigera Zehntner (=Oregma lanigera), the same aphid host as L. oregmae in the Philippines. Furthermore, apart from the drawing of the petiole of L. myzakkaiae, which doesn’t follow the original description (i.e., two lateral longitudinal wavy branched carinae), both L. myzakkaiae and L. pseudoscutellaris are morphologically similar to L. oregmae. Moreover, it seems that neither species has been mentioned in the scientific literature since their description. In our opinion, all three species are likely to be conspecific with L. oregmae. However, due the absence of type material, the authors refrained from formal synonymization

What about L. scutellaris etc.? Did you study the type series? Can you confirm the synonymy made by Stary?

The type series of L. scutellaris was not studied. However, there is not a single doubt as to whether Stary made the right decision and made it a synonym of L. oregmae

Comment 27

lines 682-683: Please check this sentence and rewrite in order it makes sense.

lines 718-720 sentence edited to: “Although L. gracilis has also been reported from Myzus [43,44], in the light of the new species discoveries, it is not clear whether the specimens reared from Myzus are indeed L. gracilis or might be some other Lipolexis species, such as L. labialis sp. n. or L. peregrinus sp. n.”

Comment 28

line 684: replace "study" by "analysis"

Replaced.

Comment 29

line 686: correspond "with" it (?)

line 725: Changed to “correspond to it”.

Comment 30

line 693: an extra space

Deleted.

Comment 31

line 696: The new sentence should not start with the abbreviated Latin name, use it in full in this case

line 735: Changed to full name.

Comment 32

line 694: "possesses...distribution" sounds odd to me (maybe "has" would be better?)

line 733: changed from “possesses” to “has”

Comment 33

line 731: "attacks some tribes" - please rewrite, it does not attack tribes but "the representatives/members of tribes..."

line 770: added “representatives” to the sentence

Comment 34

line 744 and in other places: Please, do NOT use the "possibly basal" or "basal" for the genus Ephedrus... I know what you want to say but this is not a good expression.

lines 789, 797: term basal excluded.

The same for using "primitive" characters...what does it mean? I would avoid using these terms.

Line 799: term “primitive” replaced with “plesiomorphic”

Comment 35

Authos Contributions

Some authors are not explicitly mentioned here, although I assume they all "reviewed and recised the paper" which was their role and this is why they are the co-authors and not in Acknowledgements.

All the authors are now explicitly mentioned.

Reviewer 2 Report

This study aims to revise the world-scale taxonomy of the genus Lipolexis with description of six new species. Since the paper is composed of a vast amount of data, it is estimated that the authors' hard work and accumulated competencies have been demonstrated.

This paper contains important information on the taxa, and effectively solved the species boundary problem through COI barcoding analysis. It provides morphological and ecological information on the six new species that are newly described, as well as morphological photos for each part.

However, there are a few things that need to be corrected in the manuscript, so before this paper is accepted, it must go through a simple revision procedure by the authors. Thank you.

>Minor comments were included in the review PDF file.

Author Response

Reviewer 2

We would like to thank Reviewer 2 for the constructive comments and suggestions that will improve our manuscript.

Comment 1

Did you use 'Burn in' trees before estimating? So Number of 'Burn in' trees? also need description of replications and sampling frequency on the simulation.

lines 151-155 added:  The analysis ran for ten million generations, the sampling was conducted every 1000 generations, while the first million of trees was discarded as a burn-in. The effective sample size (ESS) of the parameters of the Markov chain Monte Carlo was estimated by Tracer v1.7.1. The saturation level for the third codon position was inspected in DAMBE software.

Comment 2

version?

line 150: added “(v1.10.4)”

Comment 3

Line 151 Please describe method of bootstrapping

line 157: Added “(Tamura 3-parameter model, 2000 replicates, total of 683 positions in the final dataset)”

Comment 4

Please write the scientific name in italics

line 208: Changed to italics

Comment 5

Line 210 looks like different font compare with Mesosoma, Metasoma, ect...

Please check again

line 217: Changed to italics

Comment 6

Line 236 different font with other species.

please check again

The terms Host, Coloration, Head and Body length are changed to italics throughout the manuscript

Comment 7

Line 244 Please write the scientific name in italics

Changed to italics.

Comment 8

line 255 It would be better use: Dorsal view of petiole

Changed throughout the manuscript to: “petiole, dorsal view”

Comment 9

line 299: Please write the scientific name in italics

Changed throughout the manuscript.
